# Computing-in-memory architecture for Kolmogorov-Arnold networks based on tunable Gaussian-like memory cells

Zhixing Wen[1,2,6], Qirui Zhang[1,6], Jiangang Chen [1,6], Tianhua Yang[1], Fan Yang[1], Xuemei Wang[1], Qing Liu [1], Xiao Luo [1] ✉, Peng Lin [3], Liang-Jian Deng [4] ✉ & Fucai Liu [1,5] ✉

Inspired by the Kolmogorov-Arnold representation theorem, the Kolmogorov-Arnold networks serve as promising alternatives to multilayer perceptrons. Kolmogorov-Arnold networks utilize a superposition of finite basis functions to implement variable continuous univariate activation functions, offering greater flexibility and adaptability. However, the hardware implementation of its basis functions remains costly, making it challenging to achieve complex computations with minimal equipment. Here, we designed the device defined as Gaussian-like memory cell, composed of a Gaussian transistor and a memristor, to ensure the tunable Gaussian-like current-voltage responses. Furthermore, we constructed the circuits based on Gaussian-like memory cells to accommodate the parallel inference computation of Kolmogorov-Arnold networks. This study demonstrates that the proposed architecture based on Gaussian-like memory cells can effectively maintain the algorithmic advantages across various tasks including one-dimensional function regression, image recognition, partial differential equation solving, and time-series forecasting. Notably, the proposed architecture achieves significant improvements in energy efficiency. The results provide a promising avenue for computing-in-memory architecture for Kolmogorov-Arnold networks, and expand the flexibility and efficiency of the neuromorphic computing paradigm.

With the rapid advancement of AI, from AlphaGo[1] to ChatGPT-4o[2,3], deep neural networks (DNNs) are evolving at an incredible pace, and they have a wide range of applications in various fields, including speech recognition, visual detection, and other fields such as protein structure prediction[4,5]. Multilayer perceptrons (MLPs)[6,7] are regarded as the default frameworks for implementing nonlinear regressors in

DNNs (Fig. 1a), the importance of which can never be overstated. DNNs have evolved to contain tens to hundreds of billions of parameters[8], the demand for computational resources for MLPs is also growing rapidly. Notably, the typical computation process of MLPs involves a large number of vector-matrix multiplication (VMM) operations. However, due to the separation of memory and computational units,

[1]School of Optoelectronic Science and Engineering, University of Electronic Science and Technology of China, Chengdu, China. [2]Yangtze Delta Region Institute (Huzhou), University of Electronic Science and Technology of China, Huzhou, China. [3]College of Computer Science and Technology, Zhejiang University, Hangzhou, China. [4]School of Mathematical Sciences & Multi-Hazard Early Warning Key Laboratory of Sichuan Province, University of Electronic Science and Technology of China, Chengdu, China. [5]State Key Laboratory of Electronic Thin Films and Integrated Devices, University of Electronic Science and Technology of China, Chengdu, China. [6]These authors contributed equally: Zhixing Wen, Qirui Zhang, Jiangang Chen. ✉e-mail: luox@uestc.edu.cn; liangjian.deng@uestc.edu.cn; fucailiu@uestc.edu.cn

**Fig. 1 | Comparison of memristor-based MLPs and GMC-based KANs. a** The structure diagram of MLPs shows the calculation principle of the dense layers and the electrical characteristics of the memristors. **b** The structure diagram of KANs shows the calculation principle of learnable activation functions and transfer characteristics of GMCs. **c** Each GMC consists of a Gaussian transistor and a Gr/CIPS/Gr memristor. **d** Schematic diagram of the device structure and drain current ($I_D$) -gate voltage ($V_g$) curve of a Gaussian transistor. Sharp increases and decreases in drain current can be observed by switching between P-type and N-type operations through controlling the operating state of the gate. **e** The schematic diagram of tunable GMC is realized by stimulating and changing the conductance of the memristor to control the voltage at both ends of the Gaussian transistor, thus realizing the controllable Gaussian-like functions.

the von Neumann architecture results in substantial energy consumption and high latency for data shuffling between units for implementing VMM[9], severely limiting the computational efficiency of MLPs. In contrast, neuromorphic computing based on memristive devices with multilevel tunable resistance states offers a promising non-von Neumann computing paradigm[10], where the network parameters are stored directly, eliminating the cost of data transfer. By directly using Ohm's law for multiplication and Kirchhoff's law for accumulation, they can build parallel computing-in-memory (CIM) units that support VMM[11], improving the speed and energy efficiency of analog memory computing[12,13]. Such hardware implementations extend the integration density and computing efficiency of MLPs in utilizing huge computational resources[14].

Recently, inspired by Kolmogorov-Arnold (K-A) theorem, Liu et al. proposed the Kolmogorov-Arnold networks (KANs) as powerful alternatives to MLPs[15]. KANs replace linear weights with learnable activation functions, which are computed by combining multiple adjustable basis functions (Fig. 1b). This change offers several advantages, typically including flexibility and adaptability, allowing for the dynamic adjustment of activation functions and the ability to adapt to different data patterns[16]. Furthermore, KANs exhibit faster neural scaling laws[15], meaning that performance improves faster as the model parameters increase. More uniquely, each basis function introduces the local plasticity similar to that of human brains[17,18], enabling models to avoid catastrophic forgetting[15,16]. However, due to the demand for a large number of function superpositions, KANs incur higher complexity on conventional computing machines. In particular, the original KANs use third-order B-splines as the basis functions, which means that even a single B-spline must undergo three complex recursive operations[15,19]. In complementary metal-oxide-semiconductor (CMOS) technologies, this requires a large number of amplifiers, transistors, and chip area for operations, resulting in substantial hardware costs in the implementation of KANs[20]. Specifically, due to the inherently complex mathematical design of KANs, which relies entirely on combinations of nonlinear parameterized basis functions (such as Gaussian kernels), traditional von Neumann architectures struggle to handle such high-density computations. For instance, graphics processing units (GPUs) achieve high computational density by efficiently parallelizing simple and regular tasks, such as general matrix multiplications (GEMMs) in MLPs[21,22]. In contrast, KANs' intricate computation process and their inability to process data in batches require multiple kernel executions per forward pass, significantly increasing the CPU-GPU offloading[23] pressure and reducing resource utilization[15]. Moreover, memory bandwidth limitations and the overhead from data transfer

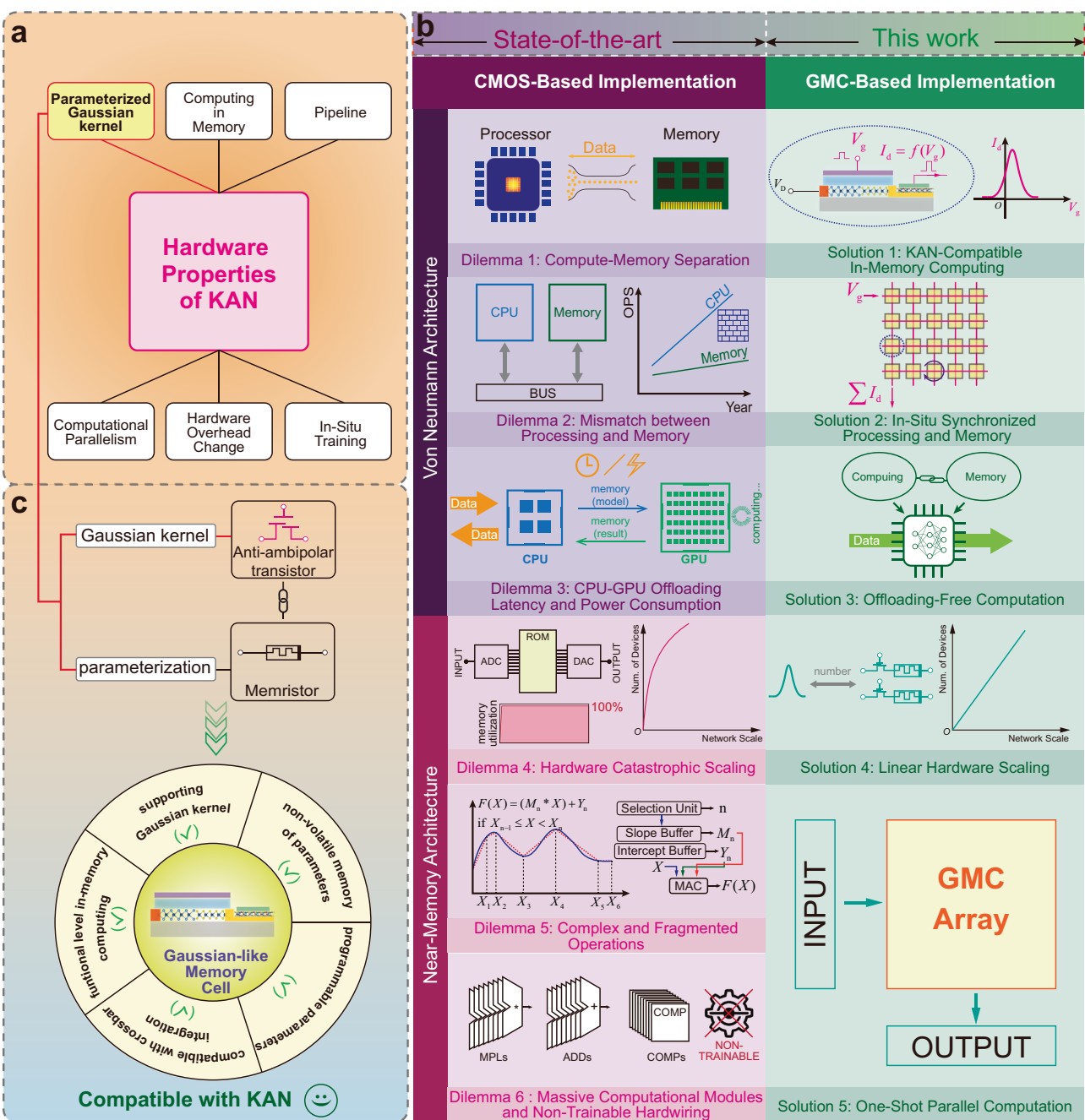

**Fig. 2 | Overview of KAN's hardware architecture strategies. a** Reference object and benchmark metrics for the hardware properties of KAN. **b** The bottlenecks encountered by the von Neumann and near-memory architectures in processing the computations of KAN, while the GMC-based architecture provides a reliable solution at the in-memory computing level. **c** The GMC ingeniously combines the Gaussian kernel provided by AAT with the parameterization capability of memristors, integrating the necessary physical foundation for the in-memory computing required by KAN.

further exacerbate power consumption and hardware resource demands[24], highlighting that KANs struggle to efficiently utilize resources within conventional architectures (the detailed explanation can be found in Supplementary Note 1).

Even for the near-memory implementations, such as application-specific integrated circuits (ASICs) based on look-up tables (LUTs)[25] or piecewise linear (PWL) approximation[26], incur significant hardware resource consumption due to the complexity of producing a single Gaussian kernel, requiring extensive circuits containing multipliers (MPLs), adders (ADDs), and comparators (COMPs). While conventional hardware can approximate Gaussian-like functions, this introduces overhead from address generation

and memory access[25]. Moreover, approaches like PWL approximation face bottlenecks with inflexibility and lack of support for dynamically learnable activation functions, as pre-stored parameters become invalid when the function updates, making such implementations unsuitable for KAN's training and one-shot multiply-accumulate (MAC) computation[27,28]. Furthermore, on the analog side, while Gaussian-like functions can be emulated using transistors[29], these implementations require continuous and precise control of voltage and current. Meanwhile, such devices are volatile and easily lose stored information upon power loss, making them unsustainable for integrated arrays, especially in low-power edge computing scenarios where long-term computation is essential.

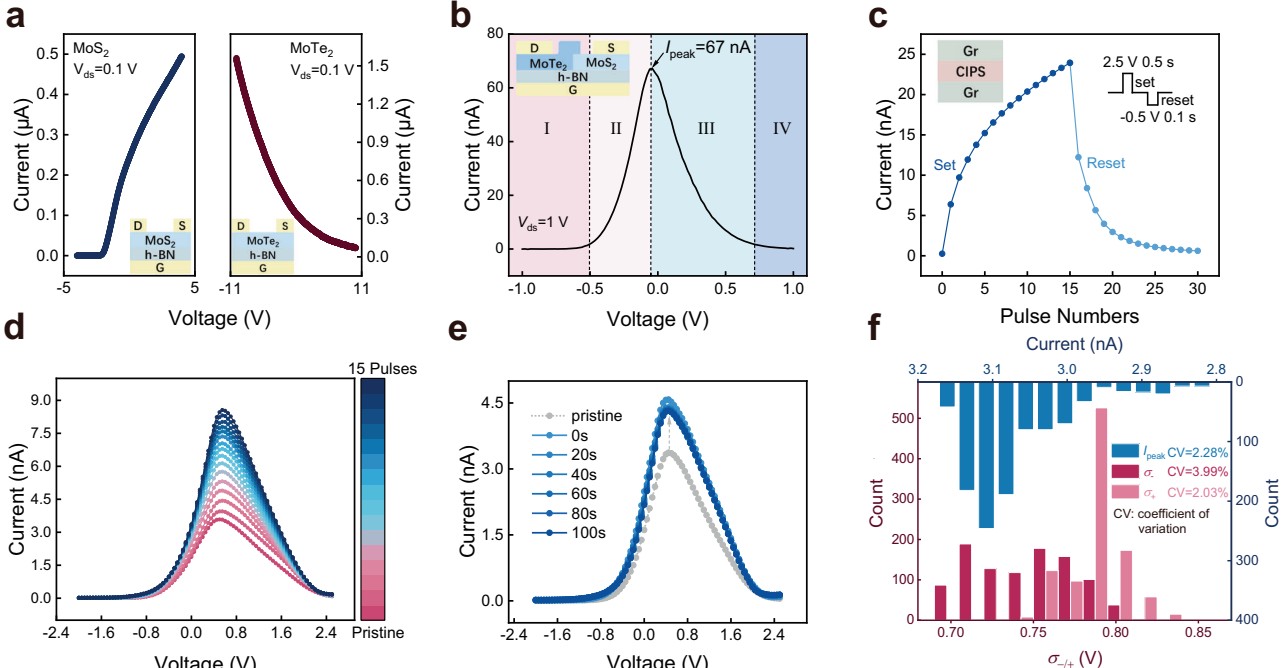

**Fig. 3 | Electrical measurements of GMCs. a** Transistor transfer characteristic curves based on independent $MoS_2$ and $MoTe_2$ channels. $MoS_2$-channel transistors exhibit NMOS characteristics, and $MoTe_2$-channel transistors exhibit PMOS characteristics. **b** The transmission curve using $V_g$ modulation junction, illustrated as schematic diagram of Gaussian transistor. **c** Potentiation and depression behavior of the Gr/CIPS/Gr memristor. **d** The transfer characteristics of GMC controlled by the write pulses (with an amplitude of 3 V and a pulse width of 0.5 s). **e** The 100 s retention performance of the GMC in pristine state (gray) after being written by two consecutive 3 V, 0.5 s pulses (blue). **f** Distributions of the Gaussian-like fitting parameters $I_{peak}$ and $\sigma_m$ extracted from the 1000 cycle endurance test.

The aforementioned facts have demonstrated that traditional hardware implementations impede the full potential of emerging algorithmic architectures like KANs. The fundamental solution lies in developing computing-in-memory devices that not only overcome the issues outlined above but also inherently align with the computational attributes of KANs. Here, each Gaussian-like memory cell (GMC) is fabricated by combining an anti-ambipolar transistor (AAT)[30] and a memristor to implement tunable Gaussian-like basis functions at the analog level, serving as a replacement for B-spline basis functions and enhancing KAN's computation performance[15,31]. This design naturally aligns with the fundamental requirements of KANs, enabling the integrated arrays to learn complex features with greater flexibility while supporting continual learning[32,33]. In electrical design, integrating GMCs into the crossbar arrays enables parallel in-memory computing tailored to KANs. Figure 2 provides an intuitive summary, presenting the details of various device properties in developmental order. In addition, these implementations are thoroughly analyzed along two aspects—reference object and benchmark metrics—and documented in Supplementary Table 1.

Given this background, the performance of GMC-based KAN (G-KAN) architecture can be validated on multiple tasks across different domains. Specifically, given that KANs have demonstrated significantly superior continual learning capabilities compared to MLPs[15,16], this study verifies that the G-KAN architecture also successfully inherits the algorithmic advantages of KANs in various AI tasks. Under this premise, the CIM advantages of G-KAN in terms of parallel computation and energy efficiency are validated, thereby demonstrating the feasibility of the architecture.

## Results
### Structure of tunable Gaussian-like memory cells
The schematic diagram of integrating a Gaussian transistor in series with a Graphene (Gr)/CuInP$_2$S$_6$ (CIPS)/Graphene memristor to construct tunable GMC is shown in Fig. 1c. In detail, we constructed Gaussian transistors using $MoS_2$ (N-type semiconductor) and $MoTe_2$ (P-type semiconductor). By controlling the gate voltage to modulate the transition between the dominant operating states of different semiconductor types, then a Gaussian-like current distribution is generated (Fig. 1d). The memristor here modulates the voltage across the Gaussian transistor through voltage division, thereby tuning the variation of a Gaussian feature, as illustrated in Fig. 1e. To verify the device performance, we first characterized the electrical transport properties of the individual $MoS_2$ and $MoTe_2$ channels. The results, as shown in Fig. 3a, reveal electron and hole-dominated transport behaviors observed in $MoS_2$ and $MoTe_2$, respectively, which are crucial for the formation of p-n junctions and Gaussian transistors. Next, we study the single-gate operation mode of the heterojunction device by independently biasing the gate of the Gaussian transistors. Figure 3b presents the transfer characteristics of the heterojunction under gate control. The device exhibits a typical Gaussian-like bipolar response relative to the control gate terminal. The $V_g$ controlled transfer curve of the junction can be divided into four regions depending on the charge accumulation profiles while changing the electric fields. Supplementary Fig. 1 schematically illustrates the band alignment capturing the doping characteristics of these four regions. In Region I, due to its p-type dominant bipolar nature, holes accumulate in the $MoTe_2$ channel, while the complete depletion of the n-type $MoS_2$ channel fully turns off the heterojunction, resulting in the lowest junction current. As the gate voltage increases, the Fermi level gradually moves upward into the bandgap, leading to a moderate decrease in hole concentration in $MoTe_2$, while the electron doping level in $MoS_2$ rapidly increases, enhancing the overall junction current (Region II). Region III refers to the stage where the depletion of the hole channel in $MoTe_2$ occurs faster than the accumulation of electrons in $MoS_2$, resulting in a decrease in junction current until the heterojunction is turned off (region IV). In addition, as shown in Supplementary Fig. 2, varying the drain voltage ($V_{ds}$) still reveals that the current of the Gaussian

transistors' electrical transfer characteristics conforms to a Gaussian-like distribution[34].

By using the memristor as a voltage divider structure, the characteristics of the GMC can be controlled. Supplementary Fig. 3 illustrates the $I-V$ curves of three cycles of an independent CIPS device, showing significant hysteresis characteristics. Under the influence of the applied electric field, $Cu^+$ ions move out of the lattice and migrate in the direction of the electric field, accumulating near the cathode electrode, which continuously affects the conductivity of the device[35,36]. When we stimulate the CIPS memristor with different write pulse counts (Fig. 3c), it can be reset to its initial state after each write pulse test, indicating the reconfigurable migration of copper ions induced by the applied electric field.

Subsequently, the electrical characteristics of the tunable GMCs are tested. In detail, Fig. 3d shows the changes in the current of the electrical transfer characteristics of the GMC under a fixed source-drain voltage after stimulated with different write pulses. As the number of write pulses increases, the peak current ($I_{peak}$) of the Gaussian curve also increases linearly, while both $V_\mu$ and $\sigma_m$ only exhibit slight variations. The Gaussian-like distribution of current is modeled by the following equation:

$$I_D = I_{peak} \exp\left[-\left(\frac{V_g - V_\mu}{\sigma_m}\right)^2\right] \tag{1}$$

where the $I_{peak}$ and $V_\mu$ respectively represent the amplitude and mean of the Gaussian-like function, while $\sigma_-$ denotes a fitting parameter modeling $I_D$ when $V_g \leq V_\mu$, and $\sigma_+$ models $I_D$ when $V_g > V_\mu$. Strong correlation between the experimental measurement data and the fitted curve is observed (Supplementary Fig. 4). These results clearly demonstrate that simple programming operations on the GMC can achieve controllable changes in the Gaussian-like distribution of the transfer characteristic current. To investigate the stability and reliability of GMCs, retention property[37] and endurance[38] evaluations are conducted. With applying two programming pulses to induce significant changes in the current, the transfer characteristic curves at 20 s, 40 s, 60 s, 80 s, and 100 s after programming illustrate virtually no changes compared to those observed immediately after programming (Fig. 3e). Figure 3f displays the distribution of characterization parameters ($I_{peak}$ and $\sigma_m$) in the Gaussian-like transfer characteristics of GMC devices under 1000-cycle test (Supplementary Fig. 5), with all results demonstrating relatively small coefficients of variation (CV). Moreover, the device-to-device uniformity is further corroborated (Supplementary Fig. 6). To further verify the learnability of GMCs, a $1 \times 2$ array is fabricated based on the characteristics of GMCs described above to represent the activation functions with continuously tunable characteristics. By applying 2, 4, and 6 pulses to each GMC, the weighted states are modulated accordingly, thereby being able to precisely control the shape of the output curve (Supplementary Fig. 7).

## Circuit design for GMC-based hardware architecture

Inspired by the network topology of KANs (Fig. 4a), this work proposes a crossbar-compatible design of GMC arrays (Fig. 4b, c). For ease of reference, 'GMC-based KAN' is termed 'G-KAN' in this paper. Such an architecture enables the inference operations of KANs to be processed in parallel at the network level. Specifically, the G-KAN's circuit integrates GMCs following the standard crossbar configuration[13,14,39]. As shown in Fig. 4b, electrical signals are conducted through densely arranged source lines (SLs), bit lines (BLs), and word lines (WLs). During computation, a constant voltage ($V_D$) is applied to all WL rows to power the devices, while each BL row delivers the gate voltage ($V_g$) encoding the input signals (Supplementary Note 2). Subsequently, the GMCs located on the same SL column accumulate currents according to Kirchoff's current law (KCL)[40]. To ensure that negative weight coefficients can be encoded while expanding the number of effective

weights (i.e., device-encoded discrete weights), the differential-pair approach is adopted[14]. Notably, the output current along the SL is first converted into a voltage by a transimpedance amplifier (TIA). Subsequently, the two output voltages of the differential pair are processed by an operational amplifier (op-amp) in the "Subtractor Configuration" to generate the final amplified output voltage (Fig. 4c, Methods). Since each edge of the KAN (i.e., the learnable activation function) is essentially realized as a linear combination of multiple basis functions (Fig. 4d), each input variable ($x$) must be propagated to multiple Gaussian-like functions. Based on the property of function translation and the mathematical relationship between the input variable and the gate voltage (Methods), the voltage amplitude to be applied to each BL can be explicitly determined (Fig. 4e). In addition, as a complementary component of the integrated circuit analysis, the scalability of the fabricated devices within crossbar arrays is further evaluated, following the fabrication theories of self-rectifying devices and employing a half-biased read scheme, as detailed in Supplementary Note 3 and Supplementary Fig. 8.

Furthermore, in terms of topological scalability, a KAN can be extended beyond the combination of basis functions by additionally configuring a residual activation ($b(x)$)[15]. Since $b(x)$ is entirely computed in the form of synapses, it can be developed in hardware by expanding with an additional memristor array (Supplementary Fig. 9). The output vector of the GMC array and the $b(x)$ array are then superimposed along the corresponding dimensions to obtain the total output of the residual-enhanced G-KAN. This extension effectively addresses the vanishing and exploding gradient problems in deep neural networks, enabling more stable training and enhanced representational capacity[41]. Notably, it should be emphasized that the residual connection serves only as an auxiliary component designed to further enhance training performance, rather than a replacement for the GMC array (see Supplementary Note 4 and Supplementary Fig. 12 for details).

To understand the potential of this neuromorphic architecture, we tested whether the G-KAN architecture can retain the algorithmic inherent property of KANs to overcome catastrophic forgetting[15,16] using a toy example of the 1D function regression task with 5 Gaussian peaks. Simply put, catastrophic forgetting poses a significant challenge in machine learning, which occurs when a neural network is trained on task 1 and then switches to be trained on task 2, causing it quickly to forget how to perform task 1[15,42,43]. Specifically, each Gaussian peak ($f(x)$, $x \in [-1, 1]$) is independently train the G-KANs (in the size of [1,1]) using 1000 data points, based on 100 basis functions per edge. During feedforward propagation, 5000 input signals evenly spaced from −1 to 1 are transmitted into the network successively, and then G-KANs output the fitted results (denoted as G-KAN($x$)). Upon learning a new peak, the model remains capable of fitting the previous peaks. After training on the final Gaussian peak, the root mean square error (RMSE) between G-KAN($x$) and the 5 peaks of the label data is merely 0.017 (Fig. 4e). Moreover, the above model is constructed and tested as a complete circuit on the Simscape Electrical platform, with detailed information provided in Supplementary Note 2 and Supplementary Fig. 13. Clearly, the G-KAN architecture can retain the algorithmic property to resist catastrophic forgetting[15,16] during the continual learning process of representing functions. Furthermore, to eliminate the concern that the continual learning capability may stem from task correlation due to overlapping training samples rather than from the network's inherent properties, the identical G-KAN mentioned above was tested on three additional non-overlapping function datasets. These evaluations eliminate the interference of task relevance and confirm the cross-functional universality of the G-KAN architecture in maintaining the continual learning capability of KANs (Supplementary Note 5 and Supplementary Fig. 14).

In terms of hardware execution efficiency, G-KANs significantly improve computational performance and reduce hardware overhead

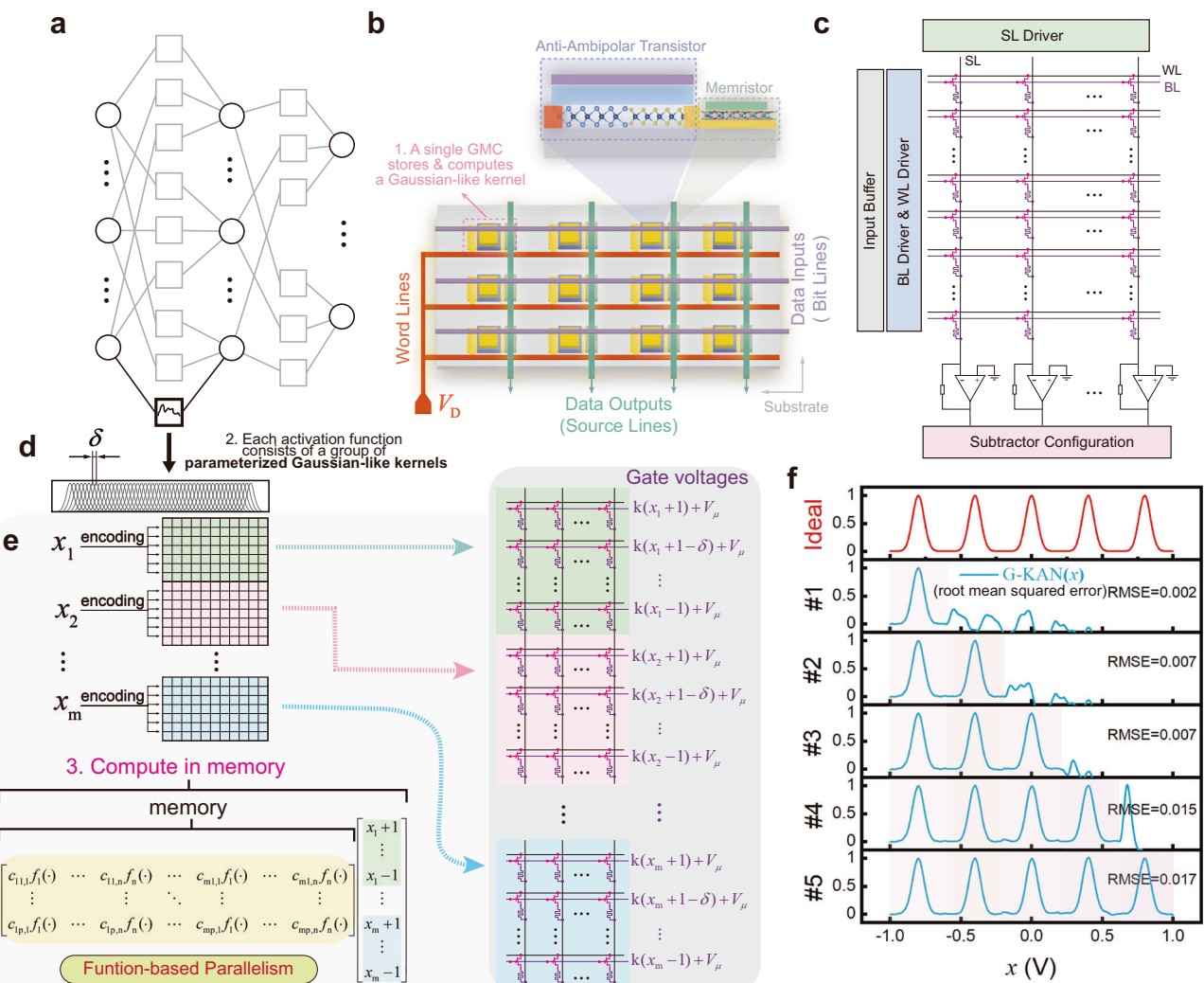

**Fig. 4 | Hardware properties of the G-KAN architecture. a, d** Topology of a KAN. Unlike MLPs, each edge is not merely a simple synapse but can be expanded into an activation function represented by a combination of Gaussian-like basis functions (Gaussian-like kernels), as illustrated in (**d**). **b** GMCs serve as the fundamental electrical units for performing tunable Gaussian-like basis functions, and are compatible with crossbar array architectures to realize analog in-memory VMM and MAC operations. **c** Schematic of the G-KAN. **e** Demonstration of function-level parallelism in a G-KAN during inference operations (m input nodes, p output nodes, with n basis functions between input-output nodes). **f** Basic performance evaluation of the G-KAN (with 100 basis functions) using a 1D function regression task as a representative example.

compared to conventional KAN architectures based on B-spline basis functions, which are typically implemented using CMOS technology[20,44] (see Supplementary Note 6 and Supplementary Fig. 15). Meanwhile, Supplementary Note 7 and Supplementary Fig. 16 demonstrate the strong robustness of G-KANs across different domains. Furthermore, it is critical to emphasize the hardware benchmark comparison with CMOS architectures for evaluating the systematic performance of CIM architectures. This study conducted a comparison between the GMC-based system and two NVIDIA GPUs. Thanks to the inherent low power consumption of GMCs and the in-situ mechanism for parallel one-shot parameterized Gaussian kernel computation and accumulation, the GMC-based system can achieve two orders of magnitude greater energy efficiency compared to GPUs. The details of the hardware benchmark statistics, GPU product models, and comparison data can be found in Supplementary Note 8.

## Further performance validation of G-KANs in various AI scenarios

Since KAN and MLP are considered fundamental building blocks supporting the skyscraper of modern deep neural networks, comparing their performance across different scenarios holds significant research value. To further explore the hardware portability of the algorithmic advantages, the G-KAN and the memristor-based MLP (M-MLP) are simultaneously trained on the same function dataset utilized in Fig. 4f to verify whether the G-KAN can retain the relative algorithmic advantages of KANs. Specifically, Supplementary Fig. 17 presents the performance of these two frameworks in the 1D function regression scenario, where the M-MLP proves to be incapable of handling the task. Even when the depth and model size of the M-MLP are substantially increased, it can at best fit the newly introduced peak while completely forgetting all previously learned peaks. Further details of this experiment can be found in Supplementary Note 5.

Function regression serves only as a basic evaluation perspective for KANs. For different input types (e.g., sequential data or image data), KANs have also been validated to exhibit strong performance in tasks such as image classification[45], PDE solving[46], and time-series forecasting[47] (Fig. 5a). Especially, performance on pattern recognition tasks serves as an important reference for evaluating deep neural networks. When handling pattern recognition tasks, the dimension of G-KANs is extended as a fully connected (FC) deep neural network

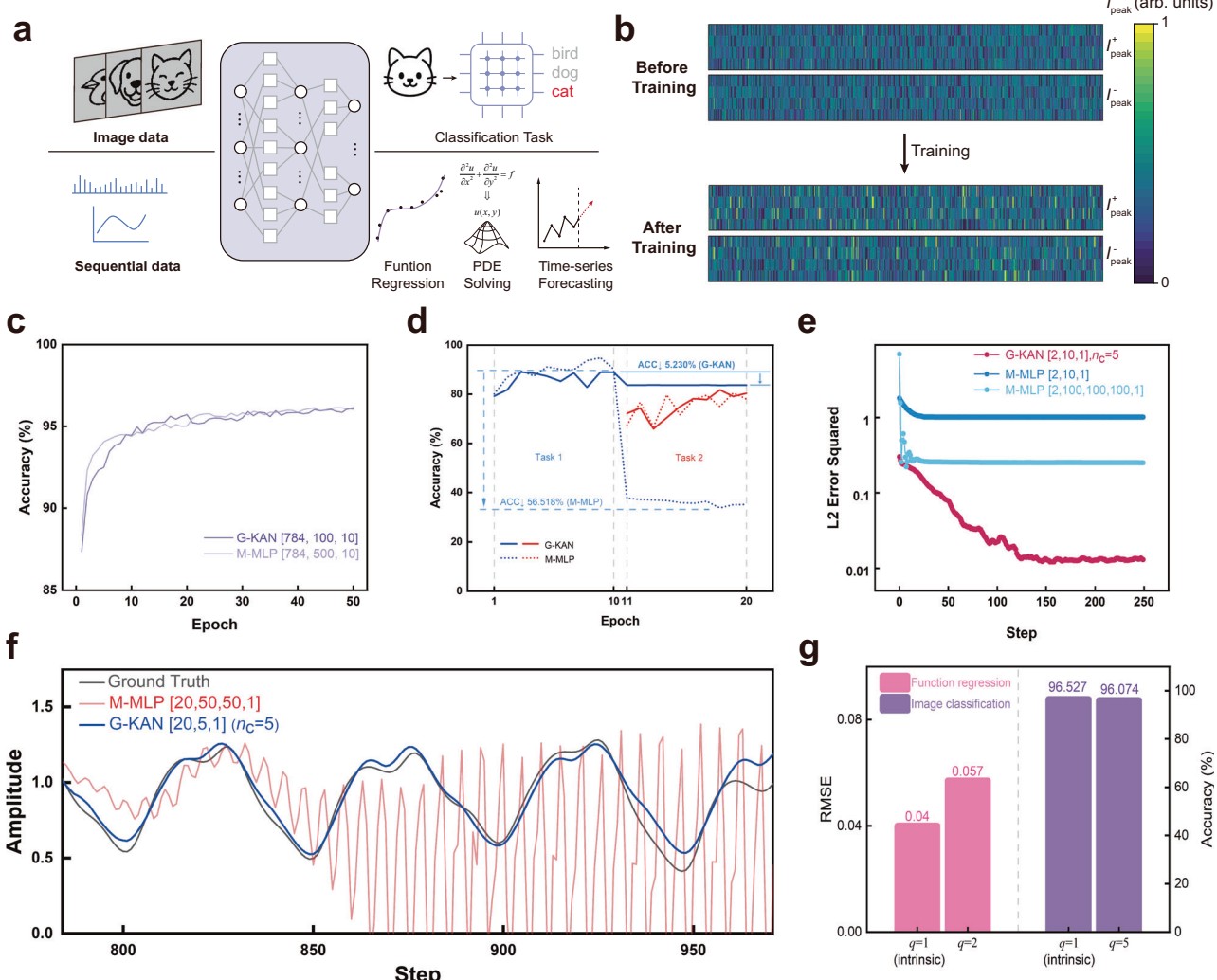

**Fig. 5 | Performance validation of G-KAN architectures across different AI scenarios. a** According to different data modalities, G-KAN can address a wide range of AI tasks. **b** Variations in the peak currents of the positive and negative components in the differential GMC pairs before and after training. The size of each heatmap is 4 × 1000. **c** The comparable learning performance of G-KAN [784, 100, 10] ($n_c$ = 4) and M-MLP [784, 500, 10] on MNIST dataset. **d** Comparison of continual learning performance for pattern recognition from MNIST (during first 10 epochs) to FMNIST (during subsequent 10 epochs). **e** Performance on PDE solving. **f** Comparison on a time-series forecasting task on the test dataset. **g** Robustness evaluation of the G-KAN architecture. The left panel presents the evaluation on the function regression task, while the right panel shows the evaluation on the image classification task.

overall. Each input image is encoded into an unfolded sequence of pixels, with each pixel intensity (i.e., input signal ($x$)) assigning the corresponding gate voltages to a GMC array in the form illustrated in Fig. 4e. Herein, the size of G-KAN is set to [784, 100, 10], with a grid size ($n_c$) of 4 for basis functions. After nearly 50 epochs (containing 60,000 images per epoch), the networks almost converged. Since the peak currents determine the shape of Gaussian-like functions, the overall state of the G-KAN is represented by the peak current heatmap of the GMC array. In this work, the peak current matrix and its variation in the final FC layer are shown in Fig. 5b, where the 4 rows represent 4 Gaussian-like functions, and the 1000 columns are derived from this layer containing 1000 edges. To compare the performance of KANs with MLPs on MNIST dataset[48], the M-MLP is preset with the identical number of training parameters for fair evaluation. The configuration mechanism for parameter count alignment is explained in detail in "Methods" section. As shown in Fig. 5c, the converged recognition accuracy of G-KAN is comparable to that of M-MLP, which demonstrates the effectiveness of G-KAN architecture in performing pattern recognition at the hardware level. Meanwhile, due to the reduced number of intermediate-layer nodes, G-KAN allows the crossbar array

to significantly reduce the number of configured SLs—by several folds —while maintaining the same count of neuromorphic devices. Taking the differential pair encoding scenario as an example, each output node essentially computes the result through double SLs[12,40]. Furthermore, additional simulation results confirm that the G-KAN architectures successfully replicate the faster neural scaling laws, which have been verified on KANs[15] (Supplementary Fig. 18).

Moreover, to validate whether the G-KAN architecture can maintain the continual learning advantage of KANs in pattern recognition scenarios, the experiment is executed by training G-KAN [784, 100, 10] ($n_c$ = 4) and M-MLP [784, 500, 10] on task 1 (the MNIST dataset) for the first 10 epochs, and then continued to train them on task 2 (FMNIST dataset[49]) for another 10 epochs. The results demonstrate that G-KANs experience less than 0.1 times the descent of accuracy compared to M-MLPs' due to the resistance to catastrophic forgetting (Fig. 5d). This indicates that the G-KAN architecture can replicate the KANs' capability in continual learning even for pattern recognition tasks. In addition, the G-KAN and M-MLP frameworks are further evaluated their performance on PDE solving and time-series forecasting tasks, with details provided in Supplementary Note 9. On the one hand, the

first task involves solving a Poisson equation with zero Dirichlet boundary conditions[15]. As shown in Fig. 5e, the L2-norm error of G-KAN is substantially lower than that of the two M-MLP architectures, even when using the same or a smaller network size. On the other hand, both frameworks are evaluated on a time-series forecasting task (on the Mackey-Glass time-series system[50]), as illustrated in Fig. 5f. It is worth noting that the outputs of G-KAN closely match the ground truth in both the training and testing stages, achieving an RMSE of only 0.071. In contrast, the M-MLP models, even with deeper architectures, exhibit good fitting performance during training but suffer from poor generalization in the testing stage. Further experimental results on the time-series forecasting task can be found in Supplementary Fig. 19. In summary, the G-KAN architectures are capable of retaining the cross-scenario performance advantages of KANs in hardware deployment.

To investigate whether G-KANs both demonstrate robustness in AI tasks involving sequential data or image data, G-KANs with sizes [1,1] ($n_c = 100$) and [784,64,10] ($n_c = 4$) are tested on function regression and image classification, respectively. To more clearly highlight the anti-interference capability, the tests are conducted under intrinsic characteristic perturbations doubled in magnitude. Here, $q$ denotes the amplification factor of the CV for each characteristic parameter. When $q = 1$, the system operates under the intrinsic characteristic perturbations of GMCs; when $q > 1$, it reflects the combined effects of external disturbances, thereby producing more severe perturbations[38]. During normal array operation, considering the drift of the sensing circuits, $q$ is expected to be only slightly greater than 1[40]. As shown in Fig. 5g, G-KANs exhibit strong robustness, which is particularly important for the scalability of array integration[49]. In addition, for the correspondence between error statistics and the actual effect on GMC arrays, the detailed analysis can be found in Supplementary Note 10.

## Discussion

In summary, we proposed a class of memory cells termed GMCs that integrate Gaussian transistors in series with memristors. This hybrid structure enables the systematic programming of Gaussian-like $I$–$V$ characteristics. Based on these cells, we designed specialized circuits and developed a CIM architecture named G-KAN. By connecting multiple GMCs and precisely tuning the output amplitude of each GMC's electrical transfer curve, G-KAN flexibly realizes various activation functions tailored for KANs. Comprehensive evaluations based on simulation indicate that the G-KAN architecture effectively replicates the inherent cross-scenario continual learning advantages of KANs, while demonstrating strong robustness during system training. Meanwhile, thanks to the low power consumption characteristics and the parallel computation capability for parameterized Gaussian functions, the energy efficiency of the GMC-based system improves two orders of magnitude compared to conventional CMOS architectures. This work establishes a practical and scalable approach to KAN's hardware architecture and lays a solid foundation for advancing next-generation neuromorphic computing architectures.

## Methods

### Fabrication and measurement of GMCs

The memory cell was achieved by a dry transfer method. Graphene, hexagonal boron nitride (h-BN), $MoTe_2$, $MoS_2$, and CIPS flakes were mechanically exfoliated onto a silicon wafer coated with 285-nm $SiO_2$. For the Gaussian transistor, using an optical microscope in combination with transfer stage at temperatures of 70–90 °C, graphene, h-BN, $MoS_2$, and $MoTe_2$ were sequentially picked up with poly-dimethylsiloxane (PDMS) covered in polycarbonate and then placed on a blank silicon wafer at 180 °C. The wafer was subsequently soaked in chloroform overnight to remove the polycarbonate layer. For the Gr/CIPS/Gr memristor, the same transfer method was used to sandwich the CIPS flake between two graphene flakes. Metal electrodes (Cr

8 nm/Au 50 nm) were then introduced onto the two parts through photolithography and electron beam evaporation. Finally, the two modules were connected using an ultrasonic wire bonder. The electric measurement of the cell was conducted using a FS-Pro 380 semiconductor parameter analyzer. In the Gaussian transistor test, $MoS_2$ ends are grounded, and voltage is applied from $MoTe_2$. For the GMCs test, one end of the memristor is grounded. Voltage is applied from $MoTe_2$. When the Gaussian-like current is regulated, a voltage is applied to the gate of the Gaussian transistor to the ON state.

### Learnable coefficients encoded by the modulable Gaussian-like transfer characteristics of GMCs

Given $G(x) = \exp[-(\frac{x}{\sigma_m})^2]$ based on the co-center transfer characteristics, the GMC essentially outputs a scaled factor of $G(x)$ (i.e., $I_{peak}G(x)$). Within a limited peak current range $[I_m, I_M]$, the quantized Gaussian-like function becomes $cG(x)$. Notably, the $c = \frac{I^+ - I^-}{I_M - I_m} \in [-1, 1]$, where the $I^+$ and $I^-$ represent the minuend and subtrahend in a differential pair, respectively. Overall, this paradigm supports differential operations based on Gaussian-like functions at a hardware level.

### Principle of MAC operations in G-KANs

Each $b(x)$-row is composed of a row of Gr/CIPS/Gr memristor differential pairs, and the synaptic weights are mapped onto the conductance of memristors. To facilitate electronic architecture, the Sigmoid linear unit (SiLU($x$))[15] in $b(x)$ is replaced with the rectified linear unit (ReLU($x$))[15] (further details can be found in Supplementary Note 11, Supplementary Fig. 20 and 21). Therefore, the MAC operation of the G-KAN with extended residual connections can be expressed by the following equation:

$$y_j = \sum_i w_b \text{ReLU}(x_i) + \sum_i \sum_{n=1}^{n_c} c_n G_n(x_i) \tag{2}$$

Herein, $x_i$ represents any neuron in a layer, where the first term in the summation is $\sum_i b(x_i)$, and the second term denotes the learnable activation functions ($\phi(x_i)$) determined by the coefficients ($c_n$). Then the neuron ($y_j$) in the next layer can be computed based on the MAC operation.

### Simulation and configuration of G-KANs

The simulation in this work is conducted based on the Python framework and Simscape Electrical platform (Supplementary Note 2). Supplementary Note 12 provides the preset configurations for model training. In addition, the training parameters updated at each iteration are mapped to the physical states of the devices (i.e., peak currents for GMCs and conductance for memristors).

For parameter count alignment, since each edge in a KAN carries $n_c$ additional training parameters compared to M-MLP, the number of parameters in G-KAN and M-MLP should satisfy the given equation:

$$P_{KAN} = (1 + n_c)P_{MLP} \tag{3}$$

Herein, $P$ represents the number of training parameters in a network. Therefore, given that G-KAN is configured with a network architecture of [784, 100, 10] ($n_c = 4$), the M-MLP with the same number of training parameters is configured as [784, 500, 10].

### Reporting summary

Further information on research design is available in the Nature Portfolio Reporting Summary linked to this article.

## Data availability

All data that support the findings of this study are available within the article and its Supplementary Information files. The MNIST dataset

utilized in this work is publicly available at https://pytorch.org/vision/main/generated/torchvision.datasets.MNIST.html, while the Fashion-MNIST (FMNIST) dataset utilized in this work is also publicly available at https://pytorch.org/vision/0.19/generated/torchvision.datasets.FashionMNIST.html.

## Code availability

Code for the simulations is publicly available at GitHub (https://github.com/VennUeSTC-NeuroComput/GMCSimu).

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

## Acknowledgements

This work was supported by the National Natural Science Foundation of China (92477115, F.L.), Sichuan Science and Technology Program (2025ZYD0182, F.L.), the National Key Research & Development Program (2020YFA0309200, F.L.), the National Natural Science Foundation of China (62274024, X.L.), and Sichuan Province Key Laboratory of Display Science and Technology.

## Author contributions

F.L. supervised the project. F.L., Q.Z., Z.W., J.C., and X.L. conceived the idea and designed the experiments. Q.Z. Fabricated the device, performed the electronic measurements, assisted by J.C. and F.Y. Z.W. conducted the circuit design and simulation, assisted by T.Y. and X.W. Q.L. prepared the single crystal. L.D. supervised the algorithm and mathematical theory. P.L. provided guidance on the circuit schematic design. F.L., Q.L., X.L., L.D., and P.L. analyzed the data. Z.W., Q.Z., J.C., and F.L. wrote the manuscript with input from all authors. All authors discussed the results.

## Competing interests

The authors declare no competing interests.
