## [Transparent Peer Review file · Nature Communications]

Computing-In-Memory Architecture for Kolmogorov-Arnold Networks Based on Tunable Gaussian-Like Memory Cells

Corresponding Author: Professor Fucai Liu

Version 0:

Reviewer comments:

Reviewer #1

(Remarks to the Author)

The paper proposed a computing-in-memory (CIM) implementation for Kolmogorov-Arnold Networks (KANs), to address KANs' efficiency issue. In place of cubic splines, the authors use Gaussian-like basis functions to parameterize univariate learnable activation functions. The Gaussian-like basis function can be achieved by a Gaussian-like memory cell (GMC), composed of a Gaussian transistor and a memristor (voltage divider structure). GMC-based KANs exhibit good performance and can avoid catastrophic forgetting in 1D function fitting and image classification. The compute-in-memory implementation is more efficient than previous KAN implementations and memristor-based multilayer perceptrons (MLPs).

This paper addresses an urgent problem in AI: efficient hardware implementation of KANs. KAN is a new type of neural network proposed in early 2024. Despite its recency, it has been cited over 500 times due to its promising algorithmic abilities (accurate, interpretable, avoiding catastrophic forgetting). However, a major limitation of KANs is the lack of efficient hardware implementation. The MAC (multiply-accumulate) calculations in KANs are different from normal VMM (vector-matrix-multiplication) computations in MLPs. GPUs are designed to speed up VMM but not MAC, so KANs call for new hardware architectures that can support efficient MAC computations. This paper, to the best of my knowledge, is the first work to design KAN hardware. Besides KANs, the proposed hardware is expected to have wide applications -- it can apply to many algorithms involving learnable activation functions or Gaussian kernels. However, I acknowledge that my main expertise is in software (algorithms), so I am unable to evaluate the novelty of GMC in the literature of electronics.

The paper's claims are well supported by their evidence. It is satisfying to see that many algorithmic advantages of KANs are demonstrated: (1) neural scaling laws (Supplementary Figure 7); (2) continual learning (Figure 3); (3) scalability to images (Figure 4).

However, there are a few algorithmic choices that seem somewhat arbitrary and should be better clarified: (1) In the 1D function fitting task, why is MAE loss used instead of the MSE loss? (2) For image classification, when switching from MNIST to Fashion MNIST, the optimizer is changed from Adam to SGD. What will happen if Adam is used for the second stage as well? (3) The comparison between KAN and MLP is subtle (this is a problem in the original KAN paper by Liu et al.), because when the network shape is kept the same, KANs may have more parameters than MLPs because of grid sizes. Figure 4c may not be a fair comparison since G-KANs have more parameters than M-MLPs. Supplementary Figure 7 is fairer since hyperparameters are swept, reflecting the performance-efficiency tradeoff. Figure 4d is fair since performance is fixed (90% accuracy). However, I did not find the corresponding shapes of KAN and MLP in Figure 4d. Also, it would be nice to have a mathematical formula, indicating how many GMCs and memristors are needed for a KAN with a given shape and a certain grid size, to show how the number of devices scales against the size of the KAN network.

Minor comments: (1) in the abstract, 'a combination of finite basis functions', do you mean 'finite element'? (2) 'variable univariate activation functions', 'variable' is redundant? (3) I think 'symbolic regression' is not a correct word (symbolic regression means extracting out symbolic formulas of numeric data), instead using 'function fitting', 'function regression', 'supervised learning' or 'curve fitting', etc.

Reviewer #2

(Remarks to the Author)

While the manuscript proposes a hardware implementation of Kolmogorov-Arnold Networks (KANs) using Gaussian-like memory cells (GMCs), there are several significant concerns that, in my opinion, prevent it from being accepted for publication in Nature Communications.

1. The performance (higher accuracy and overcoming catastrophic forgetting) demonstrated in the manuscript primarily stems from the advantages of implementing neural networks using KAN. However, the manuscript does not sufficiently highlight the benefits of using GMC to implement KAN.
2. The manuscript lacks a comparative analysis between the proposed GMC-based KAN and a conventional CMOS-based KAN implemented on electronic computers, particularly in terms of key performance metrics such as energy consumption, device volume, and data throughput.
3. While simulations are used to illustrate the advantages of the proposed approach, they cannot serve as strong evidence, especially for a physical hardware system. There is often a significant gap between simulations and physical experiments, making the reliance on simulations insufficient to support the claims.
4. The manuscript does not sufficiently address whether the scalability and stability of the GMC array can support a larger KAN for more complex intelligence tasks. A further theoretical analysis of the scalability limits of the GMC array would be valuable.
5. The physical experiments presented in the manuscript focus solely on the performance of a single device. However, the manuscript does not discuss whether the proposed GMC array relies on consistency between different units and whether this could lead to additional hardware overhead. There is a lack of relevant discussion regarding the robustness of the proposed structure.
6. The GMC-based KAN proposed in the manuscript is based on real-world analog devices. However, any analog device inevitably introduces errors, especially in large-scale arrays, where the impact of errors becomes more significant. The manuscript lacks a discussion and analysis of these errors, which raises doubts about the feasibility of the proposed architecture.
7. In the experiment validating the continual learning capability of the GMC-based KAN, the presented 1D symbolic regression task does not provide strong evidence. Factors such as task correlation and potential parameter redundancy may interfere with the conclusions. The study lacks comparative experiments or continual learning experiments across different 1D regression tasks to make the conclusion more convincing.
8. In the comparison between KAN and MLP, the parameter counts corresponding to the same network size (764-64-10) for KAN and MLP are not comparable (KAN has more parameters). The specific details of the network configurations are not presented to demonstrate the fairness of the comparison. To ensure a fair comparison, more details about the network structures and parameter settings should be provided.
9. In the MNIST handwritten digit classification task, as shown in the training curves (Fig. 4c and 4e), it appears that neither the GAN nor the MLP training process has fully converged. Therefore, such results cannot rigorously demonstrate the performance differences between the two methods.
10. The symbol regression and MNIST image classification tasks presented in the manuscript have shown the basic intelligence computing capacity for the GMC-based KAN. However, these tasks could also be well-handled by conventional MLP or CNN networks. To prove the advances of the proposed physical KAN, extra tests on more complex tasks e.g., CIFAR or ImageNet tasks will be appreciated.

In conclusion, the unclear advantages of the proposed method and the lack of sufficient experimental validation significantly limit its potential to make a meaningful contribution to the field. Therefore, I do not recommend its publication in NC.

Reviewer #3

(Remarks to the Author)

The manuscript proposes a Gaussian-like memory cell (GMC), consisting of a Gaussian transistor and a memristor, as the basis function for Kolmogorov-Arnold Networks (KAN), an emerging neural network algorithm. While exploring efficient hardware implementations of KAN is an important topic, this manuscript only scratches the surface, relying on pure simulation data and very limited experimental results from a single cell. Further analysis and more comprehensive studies are necessary for this manuscript to be considered for publication. Detailed comments are as follows:

1. The core contribution of this work is the development of the GMC. However, the manuscript only provides minimal experimental data, such as the transfer curve of the memory cell in Fig. 2. Additional device characterization, such as endurance testing etc., is recommended to demonstrate the reliability and learning capability of the proposed memory cell. In other words, what are the full suite of device properties that are required to run the KAN algorithm?
2. The circuit design presented in Fig. 3 is more of a functional block representation than a detailed circuit implementation.

To adequately demonstrate the circuit-level design, the manuscript should include explicit circuit diagrams for the key blocks rather than just describing their functions. More importantly, a circuit capable of running the KAN should be experimentally built and tested.

3. Are the results shown in Fig. 3e from experimental setups or pure simulations? If they are based on experimental setups, a photo of the memory cell array should be included to support the findings. The manuscript should provide clear details on the simulation setup and parameters if they are from simulations.

4. The claim regarding “continual learning resilience against catastrophic forgetting” in Fig. 3 appears to be an inherent benefit of the KAN rather than the GMC. To substantiate this claim, a comparative analysis should be conducted by evaluating the performance of the same KAN structure using the GMC versus other basis functions. Otherwise, this claim only reiterates the advantages of KAN without demonstrating the specific contributions of the GMC.

5. Similarly, beyond simply demonstrating that G-KAN can perform classification tasks like KAN, the manuscript should include studies on the trade-offs introduced using the GMC. Specifically, the impact of computing noise from the analog device on the performance of G-KAN should be evaluated, as well as its generalization capability to other tasks when using only a Gaussian-like function as the basis function.

Version 1:

Reviewer comments:

Reviewer #1

(Remarks to the Author)

I'm glad to see that authors have addressed all my concerns.

Reviewer #2

(Remarks to the Author)

The revised manuscript provides more comprehensive details regarding the implementation of the proposed G-KAN and addresses several of my previous concerns. However, there are still some important issues that need to be addressed.

1. While the response includes a comparison between G-KAN and M-MLP to demonstrate performance advantages, these benefits appear to stem largely from the inherent characteristics of the KAN architecture itself. The revised manuscript still provides limited discussion on the specific advantages of the GMC-based implementation. More thorough discussion and comparative visual evidence are required to underscore the contribution of the proposed GMC implementation.

2. The experiment involving two GMC units in the revised manuscript only demonstrates the functional feasibility. However, it does not analyze the potential errors or scalability issues that may arise when integrating multiple units. As such, it is insufficient to convincingly support the practical feasibility and scalability of a full GMC array. Further experimental results and in-depth analysis are necessary to substantiate this claim.

3. The response includes an error modeling of device variability and analyzes its impact on task performance. However, it remains unclear whether the modeling aligns with actual physical behavior. To justify the validity of this approach, experimental validation using measured device variability is necessary.

4. The revised manuscript states that the RMSE of G-KAN remains below 0.4 when the factor q reaches 50. However, it is not evident what level of physical error this corresponds to in practice. Is such a quantization level sufficient in a real-world hardware scenario? A more detailed explanation is needed to substantiate the robustness of G-KAN under practical conditions.

5. The response mentions methods that allow the weighted coefficients of cells to be dynamically adjusted during training to compensate for effects such as current drift caused by device variations. However, these methods are designed primarily for conventional CNN or MLP architectures, which are fundamentally different from the proposed G-KAN. Whether such techniques are applicable to G-KAN requires further clarification.

Reviewer #3

(Remarks to the Author)

The authors have conducted additional experiments and simulations that partially address my initial concerns regarding device measurements. However, the circuit designs presented in this manuscript remain underdeveloped and are insufficient to support even a design-level demonstration of computing-in-memory (CIM) for Kolmogorov–Arnold Networks (KAN), let alone an implementation of CIM-based KAN. Several significant concerns remain, even after the revisions, and the manuscript does not meet the standards required for publication in Nature Communications.

1. The circuit designs presented in this manuscript are difficult to understand and appear unrealistic, which significantly undermines the authors' claim of implementing KAN using a CIM approach. The poor quality of the circuit design raises serious concerns about the credibility of the simulation results. Below are several fundamental questions that support this assessment:

1) What do the triangle symbols represent in inputs and outputs of Fig. 3a? What are their functions?

- 2) What are the triangle symbols in Fig. 3b? Are they the same components as those in Fig. 3a?
 - 3) Fig. 1c shows that a GMC consists of a Gaussian transistor and a memristor. However, in Fig. 3b, memristors appear as independent devices when forming differential pairs. How is this structural change justified? What are the characteristics of these independent memristor differential pairs?
 - 4) In Fig. 3c, what are the triangles with one terminal connected to x ? Where are the other terminals connected?
 - 5) In Supplementary Fig. 8, what is the 1T1R architecture, and is it the same as a GMC cell? The diagram includes numerous switches, what are their specific functions? Moreover, what is the difference between these switches and the MUXes, given that they are typically considered equivalent in circuit design?
 - 6) In Supplementary Fig. 9, a ReLU function cannot be implemented with only a single analog switch. How is the comparison operation (i.e., determining whether $x < 0$ or $x > 0$) realized in the analog domain?
 - 7) Simulink is generally used for system-level simulations, and VHDL is designed for modeling digital circuits at the behavioral level. Since the proposed CIM framework targets fully analog circuits, the simulation results shown in Supplementary Fig. 10 are not representative of analog behavior and are therefore unconvincing.
2. The claimed advantages of implementing KANs using GMC-based CIM are not supported by solid evidence. The algorithm-level benefits highlighted by the authors are inherent to the KAN architecture itself and are not specific to the proposed hardware implementation. At the hardware level, the claim of reduced device count compared to MLPs is based on an unfair comparison: the authors only consider the number of memristor devices required for KANs versus MLPs, while overlooking the significantly higher circuit complexity associated with implementing KANs using GMCs. This undermines the validity of the claimed hardware efficiency.

Version 2:

Reviewer comments:

Reviewer #2

(Remarks to the Author)

I appreciate the authors' comprehensive responses, which have satisfactorily addressed all my previous concerns. Therefore, I recommend the manuscript for publication.

Reviewer #3

(Remarks to the Author)

The authors have thoroughly revised both the manuscript and the supplementary information, addressing most of my previous concerns regarding peripheral circuit designs. While the manuscript has improved in technical details, I still have concerns about the presented comparisons and several major claims. As I understand it, the primary contributions of this work are the fabrication and characterization of the GMC cell, along with circuit design and simulation demonstrating the feasibility of using the GMC cell for hardware implementation of KAN. However, the manuscript attributes certain learning capabilities, inherent to the KAN architecture itself, to the GMC cell. This attribution is not well justified and, in my view, overstates the role of the device. Some detailed comments are as follows:

1. The authors claim that the GMC-based KAN exhibits "strong learning capabilities" and "excellent continual learning capability." These properties are inherent to the KAN architecture and would be expected regardless of whether the GMC cell is used. I therefore recommend that the authors clearly distinguish algorithmic capabilities from device-level contributions. In particular, the manuscript should emphasize the advantages introduced by the GMC cell, such as improved energy efficiency and computational speed enabled by in-memory computing, rather than implying that these learning capabilities originate from the GMC cells.
2. The performance comparison between G-KAN and M-MLP in Fig. 5 is not directly relevant to the main contributions of the work. As I understand it, a comparison between KAN and MLP would yield similar trends, since the observed performance differences are attributable to the underlying network architectures rather than the use of the GMC cell. Therefore, the advantages shown in Fig. 5 do not arise from the proposed device or circuit designs. Instead, I would expect quantitative comparisons that directly demonstrate the benefits of the GMC-based in-memory computing architecture, such as improvements in energy efficiency and computational throughput, compared to a CMOS-based implementation, in order to substantiate the advantages presented in Fig. 2.
3. The manuscript primarily focuses on single-device-level experiments and circuit-level design and simulations. I therefore suggest that the authors avoid using the term "implementation" in the title and clearly clarify in the main text that the proposed G-KAN has not been "implemented" at the system level to prevent overclaim of the experimental scope and ensure that the statements accurately reflect the demonstrated results.

Version 3:

Reviewer comments:

Reviewer #3

(Remarks to the Author)

The authors have addressed my previous concerns. One additional comment is that the authors should clearly state that the

energy-efficiency estimation added in the revised manuscript is based on idealized assumptions, as the analysis considers only the G-KAN implementation, whereas the GPUs are general-purpose computing platforms.

Point-to-point response to reviewer's comments

Response to Reviewer 1's comments

General comment: The paper proposed a computing-in-memory (CIM) implementation for Kolmogorov-Arnold Networks (KANs), to address KANs' efficiency issue. In place of cubic splines, the authors use Gaussian-like basis functions to parameterize univariate learnable activation functions. The Gaussian-like basis function can be achieved by a Gaussian-like memory cell (GMC), composed of a Gaussian transistor and a memristor (voltage divider structure). GMC-based KANs exhibit good performance and can avoid catastrophic forgetting in 1D function fitting and image classification. The compute-in-memory implementation is more efficient than previous KAN implementations and memristor-based multilayer perceptrons (MLPs).

This paper addresses an urgent problem in AI: efficient hardware implementation of KANs. KAN is a new type of neural network proposed in early 2024. Despite its recency, it has been cited over 500 times due to its promising algorithmic abilities (accurate, interpretable, avoiding catastrophic forgetting). However, a major limitation of KANs is the lack of efficient hardware implementation. The MAC (multiply-accumulate) calculations in KANs are different from normal VMM (vector-matrix-multiplication) computations in MLPs. GPUs are designed to speed up VMM but not MAC, so KANs call for new hardware architectures that can support efficient MAC computations. This paper, to the best of my knowledge, is the first work to design KAN hardwares. Besides KANs, the proposed hardware is expected to have wide applications

-- it can apply to many algorithms involving learnable activation functions or Gaussian kernels. However, I acknowledge that my main expertise is in software (algorithms), so I am unable to evaluate the novelty of GMC in the literature of electronics.

The paper's claims are well supported by their evidence. It is satisfying to see that many algorithmic advantages of KANs are demonstrated: (1) neural scaling laws (Supplementary Figure 7); (2) continual learning (Figure 3); (3) scalability to images (Figure 4).

Response: We sincerely thank you for your constructive comments on our manuscript and are very pleased that the work has made a positive impression. In particular, your insights regarding the architectural challenges and the broader relevance of the proposed solution have deeply inspired us. We have carefully revised the manuscript in accordance with your suggestions, including conducting more experiments, enhancing the quantitative analysis, and presenting clearer comparative results. We hope these improvements further demonstrate the potential of the GMC-based KAN framework.

Comment 1-1: In the 1D function fitting task, why is MAE loss used instead of the MSE loss?

Response 1-1: Thank you for your insightful comments. We have compared the performance of MSE and MAE losses during training process using the root mean squared error (RMSE, the standard evaluation metric for function fitting) as the evaluation metric (arXiv: 2404.19756 (2024)). Experimental results demonstrated that

utilizing MAE loss yields slightly better fitting performance. In contrast, MSE loss tends to cause more pronounced local oscillations at the junctions between Gaussian peaks (Fig. R1).

The unstable convergence behavior of MSE stems from its mathematical formulation (MSE is the mean of the sum of squared errors): as prediction errors approach zero (Fig. R2), the squaring operation causes gradients to diminish rapidly, potentially leading to vanishing gradients and training instability (*PeerJ Comput. Sci.* 7, e623 (2021)). In contrast, MAE provides a constant, non-zero gradient almost everywhere (except at zero error) (Fig. R3), promoting more stable optimization where the errors close to zero between Gaussian peaks.

Figure R1 | Function fitting performance of G-KAN with MSE and MAE loss functions.

Figure R2 | Visualization of the absolute errors (Abs. Errors) between the KAN outputs and the label data for each segment in the Gaussian peak dataset scenario. The red vertical lines indicate the junctions between adjacent peaks, while the cyan boxes highlight regions where the prediction error is very small (error ≈ 0).

Figure R3 | Visualization of the MSE and MAE loss functions. The point where each curve touches (for MSE curve) or intersects (for MAE curve) the line (error = 0) corresponds to distinct gradient behaviors for the respective loss functions.

Comment 1-2: The comparison between KAN and MLP is subtle (this is a problem in the original KAN paper by Liu et al.), because when the network shape is kept the same, KANs may have more parameters than MLPs because of grid sizes. Figure 4c may not be a fair comparison since G-KANs have more parameters than M-MLPs. Supplementary Figure 7 is fairer since hyperparameters are swept, reflecting the performance-efficiency tradeoff.

Response 1-2: Thank you for your valuable comment regarding the fair comparison between KAN and MLP. For parameter count alignment, since each edge in a KAN carries n_c additional training parameters compared to MLP (arXiv: 2404.19756 (2024)), the number of training parameters in G-KAN and M-MLP should satisfy the given equation:

$$P_{\text{KAN}} = (1 + n_c)P_{\text{MLP}} \quad (\text{R1})$$

Herein, P represents the number of training parameters in a network, and n_c is the number of Gaussian-like basis functions contained in each edge of the KAN (i.e., the grid size).

Given that G-KAN is configured with a network architecture of [784, 100, 10] ($n_c = 4$), and M-MLP with the same number of training parameters is configured as [784, 500, 10]. Based on these presets, as shown in Fig. R4, the converged recognition accuracy of G-KAN is comparable to that of M-MLP, which demonstrates the effectiveness of G-KAN as a novel AI framework at the hardware level. Excitingly, due to the reduced number of intermediate-layer nodes, G-KAN allows the crossbar array

to significantly reduce the number of configured source lines (SLs)—by several folds—while maintaining the same number of neuromorphic devices. Taking the differential pair encoding scenario as an example, each output node essentially computes the result through two SLs (*Nature* **577**(7792), 641-646 (2020); *Nat. Commun.* **9**, 2385 (2018)). This demonstrates that G-KAN can substantially reduce the footprint of neuromorphic computing chips. The combination with our proposed hardware foundation—implementing KANs in analog CIM systems—opens new potential for advancing the practical capabilities of this promising architecture. To further explore the advantageous applications of G-KANs, we compared the performance of G-KANs and M-MLPs in additional scenarios. Specifically, we successfully validated the advantages of G-KANs on PDE and time-series tasks. Detailed information can be found in the response to Comment 2-10.

Figure R4 | The learning performance of G-KAN ([784, 100, 10], $n_c = 4$) is comparable to that of M-MLP [784, 500, 10] (equal training parameters) on image classification task.

We have added this discussion to the revised main text (on page 12 and 17). Fig.

R4 has been included as Fig. 4c in the revised main text.

Comment 1-3: For image classification, when switching from MNIST to Fashion MNIST, the optimizer is changed from Adam to SGD. What will happen if Adam is used for the second stage as well?

Response 1-3: We appreciate your attention to this detail. To clarify, there was no optimizer switching when transitioning from MNIST to Fashion-MNIST in our experiments. The confusion may have arisen from the original phrasing in the Supplementary Information, which we have revised for clarity.

Your question motivated us to conduct additional experiments using the AdamW optimizer for both datasets under settings with identical training parameter counts. Specifically, we compared G-KAN ([784, 100, 10], $n_c = 4$) and M-MLP ([784, 500, 10]) trained with SGD or AdamW in continual learning scenarios. When using AdamW as the optimizer, G-KAN showed little impairment in accuracies on task 1 after task switching, but failed to learn the data from task 2—the accuracy remained at 10%, without showing any improvement during training (see the lower panel of task 2 in Fig. R5). Meanwhile, M-MLP achieved over 60% accuracy on task 2, yet its performance on task 1 plummeted to 10% accuracy after switching tasks, indicating severe catastrophic forgetting.

Above results indicate that AdamW is not suitable as an optimizer for continual learning in these frameworks, and further support the choice of using SGD in this scenario.

Figure R5 | Comparison of continual learning performance for pattern recognition from MNIST (during first 10 epochs) to FMNIST (during subsequent 10 epochs) between G-KAN ([784, 100, 10], $n_c = 4$) and M-MLP [784, 500, 10] with AdamW optimizer.

Comment 1-4: Figure 4d is fair since performance is fixed (90% accuracy). However, I did not find the corresponding shapes of KAN and MLP in Figure 4d.

Response 1-4: We appreciate your attention to details, and your feedback helps us to improve the quality and clarity of our work. For Fig. 4d in original manuscript, the required size for G-KAN is [784, 22, 10], while the required size for M-MLP is [784, 144, 10]. The specific calculation and corresponding model shapes are provided in Fig. R6.

Figure R6 | Mathematical relationship between the number of devices and the network topology.

Comment 1-5: Also, it would be nice to have a mathematical formula, indicating how many GMCs and memristors are needed for a KAN with a given shape and a certain grid size, to show how the number of devices scales against the size of the KAN network.

Response 1-5: Thank you for raising this important point, and we elaborate more about this detail. Specifically, the size of G-KAN has the following relationship with the number of consumed devices:

$$C_{\text{device}} = C_{\text{GMC}} + C_{\text{memristor}} = 2(1 + n_c)S \quad (\text{R2})$$

Herein, S denotes the total amount of edges in the G-KAN, while C_{device} represents the cost of devices in total. Notably, $C_{\text{GMC}} = 2n_c S$ and $C_{\text{MEM}} = 2S$ denote the costs of GMCs and memristors, respectively. The factor ‘2’ comes from the use of

differential pairs—each learnable parameter is implemented using 2 physical devices.

We have added the both contents in response to Comment 1-4 and 1-5 to Supplementary Note 7 and Supplementary Fig. 17. And also referenced it on page 12 to the revised main text.

Comments 1-6: Minor comments: (i) in the abstract, 'a combination of finite basis functions', do you mean 'finite element'? (ii) 'variable univariate activation functions', 'variable' is redundant? (iii) I think 'symbolic regression' is not a correct word (symbolic regression means extracting out symbolic formulas of numeric data), instead using 'function fitting', 'function regression', 'supervised learning' or 'curve fitting', etc.

Response 1-6: We appreciate your detailed comments and address them point by point below:

(i) These two concepts differ significantly in both mathematics and application. 'A combination of a finite number of basis functions' stems from the Kolmogorov-Arnold representation theorem (KART), which states that any multivariate function can be expressed as a superposition of a finite number of continuous univariate functions (arXiv: 2407.11075 (2024)). This contrasts fundamentally with finite element methods, as KART's finite basis requirement ensures feasibility for hardware implementation, whereas infinite bases are impractical. Thus, the term 'finite' here highlights the practicality of implementing KANs in hardware.

(ii) The term 'variable' here is not redundant but serves to emphasize the contrast

with fixed activation functions in MLPs. In KANs, since the coefficients of basis functions are trainable, the resulting activation functions can theoretically approximate any shape. This inherent flexibility distinguishes KANs from MLPs.

(iii) We appreciate your valuable suggestion. After discussion, we have modified ‘symbolic regression’ to ‘function regression’ throughout the revised main text and Supplementary Information.

Response to Reviewer 2's comments

General comment: While the manuscript proposes a hardware implementation of Kolmogorov-Arnold Networks (KANs) using Gaussian-like memory cells (GMCs), there are several significant concerns that, in my opinion, prevent it from being accepted for publication in Nature Communications.

Response: We sincerely thank the reviewer for the assessment and for pointing out several important concerns. We understand the high standards of *Nature Communications*, and we acknowledge that rigorous evaluation is essential to ensure the scientific robustness and impact of published work. In response to the reviewer's concerns, we have carefully revised the manuscript and addressed each issue point-by-point (see detailed responses below). Specifically, we have strengthened the experimental evidence supporting the stability of the proposed cells. In addition, based on further simulations with varying levels of physical noise arising from non-ideal variability, we analyzed the impact of device variability on robustness. We believe these additions significantly enhance the clarity, depth, and overall scientific contribution of our work.

There is no doubt that exploring efficient hardware implementations of KANs is a critical and urgent challenge in AI, as noted by the other two reviewers. However, this area is still in its early stages of development. Therefore, our proposed architecture—based on programmable memristors—represents a significant step forward. It offers notable advantages in terms of power efficiency, robustness, and error tolerance. We

hope these revisions and clarifications help address the reviewer's concerns and effectively demonstrate the potential of our hardware-based KAN implementation to make a meaningful contribution to the field.

Comment 2-1: The performance (higher accuracy and overcoming catastrophic forgetting) demonstrated in the manuscript primarily stems from the advantages of implementing neural networks using KAN. However, the manuscript does not sufficiently highlight the benefits of using GMC to implement KAN.

Response 2-1: Thank you for your valuable suggestion, as it gives us an opportunity to further elaborate on the novelty of using GMCs to implement KAN. Currently, parallel hardware accelerators, such as GPUs, are not optimized for the spline function computations central to KAN, limiting its scalability and practical deployment. To fully unlock KAN's potential for improved accuracy and resistance to catastrophic forgetting, a hardware-software co-design approach is essential. To this end, we leveraged the learnability of memristors to propose a solution that utilizes GMCs for efficient hardware acceleration of physical KAN.

The original KAN employs B-spline basis functions, which require third-order Cox-De Boor recursive computations (arXiv: 2404.19756 (2024)), resulting in high computational and energy costs. In contrast, GMC directly generates Gaussian-like basis functions (serving as a faster alternative basis function to B-spline for computation), greatly reducing computational complexity. Moreover, replacing B-splines with Gaussians at the same model size can significantly improve forward

propagation efficiency. In 1D function regression and pattern recognition tasks, this substitution achieved a speedup of $2.073\times$ and $2.166\times$, respectively, as shown in Fig. R7.

Although previous studies have proposed using lookup tables (LUTs) in digital circuits to approximate B-spline functions and reduced recursive operations (*Proceedings of the 30th Asia and South Pacific Design Automation Conference* 693-699 (2025); *2024 Twelfth International Symposium on Computing and Networking Workshops (CANDARW)* 110-116 (2024)), LUTs suffer from severe scalability limitations. Their size increases exponentially with the number of classes, causing a sharp increase in the required number of transistors, and severely limiting the scalability of physical KAN. Although implementing B-spline functions with analog circuits offers certain advantages in resource utilization and potential scalability (arXiv: 2502.01489 (2025)), both analog and digital designs generally rely on physically separated computing and memory architectures, fundamentally limiting computational efficiency.

In comparison, GMCs inherently generate Gaussian-like functions, reducing the complexity of generating basis functions, while enabling computing in memory. More importantly, GMC's inherent programmability and in-situ computation capabilities, are well-suited for physical KAN's implementations. Table R1 provides a direct comparison of hardware resources, clearly demonstrating that employing GMCs in building efficient physical KAN systems allow to significantly reduce hardware usage and energy consumption.

Figure R7 | Statistics on the time consumption of basis functions based on GMCs or B-splines in KANs.

Table R1 | Comparison of different ways to implement spline functions

Ref.	Mode	function	Amp. adjustability	Memory	Num. of devices	Area (mm ²)	Power (μW)
Nakashima et al.	digital	B-Spline	No	No	LUTs* (96)	-	-
Yu et al.	digital	B-Spline	No	No	LUTs* (96)	-	-
Lozano Duarte et al.	digital	B-Spline	No	No	-	9.111	266.735
Lozano Duarte et al.	analog	B-Spline	No	No	42	0.073	238.5
This work	analog	Gaussian	Yes	Yes	2	2.2×10⁻⁴	~ 10⁻³

* The transistor count estimation for each 4-input LUT (4-LUT) is based on the architecture of Xilinx Virtex-4 FPGAs, where each 4-LUT can be configured as a 16-bit distributed RAM. Assuming each SRAM bit cell comprises six transistors, the total transistor count per 4-LUT can be approximated by multiplying the number of SRAM

cells (16) by six, resulting in approximately 96 transistors per LUT.

This discussion has been added to the revised main text on page 10, Supplementary Note 5, and Fig. R7 has been included as Supplementary Fig. 13c in the revised Supplementary Information. Table R1 has been included as the Supplementary Table 2.

Comment 2-2: The manuscript lacks a comparative analysis between the proposed GMC-based KAN and a conventional CMOS-based KAN implemented on electronic computers, particularly in terms of key performance metrics such as energy consumption, device volume, and data throughput.

Response 2-2: Thank you for raising this important point. We have identified three prior studies on the CMOS-based hardware implementation of KAN, which are summarized in Comment 2-1 and Table R1. All of these prior studies, regardless of whether they are based on digital or analog CMOS design, rely on a memory–computation separated architecture to implement spline functions, which introduces significant data movement overhead and thus fundamentally limits the efficiency of physical KAN implementations.

Lozano Duarte *et al.* reported that the digital CMOS implementation consumed 266.735 μW of power and occupied an area of 9.111 mm^2 , whereas the analog CMOS version consumed 238.5 μW and required 0.073 mm^2 (arXiv: 2502.01489 (2025)). In contrast, our proposed GMC achieves a significantly lower power consumption on the order of nanowatts (calculated as operating multiplication of voltage and current), with

an area of only 2.2×10^{-4} mm², thereby demonstrating advantages in both energy efficiency and area.

This discussion has been added to the revised main text on page 10 and Supplementary Note 5 in the revised Supplementary Information.

Comment 2-3: While simulations are used to illustrate the advantages of the proposed approach, they cannot serve as strong evidence, especially for a physical hardware system. There is often a significant gap between simulations and physical experiments, making the reliance on simulations insufficient to support the claims.

Response 2-3: We appreciate your valuable feedback regarding the need for experimental validation, and we fully agree that there is a gap between simulations and physical experiments. While physical verification is essential in circuit design, simulation remains an indispensable tool for demonstrating theoretical feasibility and guiding future hardware development. In this work, we aim to present a hardware-software co-design framework that can motivate further experimental studies.

For the hardware implementation of G-KAN, we have conducted comprehensive device verification and mechanism analysis (Fig. 2, revised manuscript), and further illustrated the circuit design (Supplementary Note 1 and Fig. 8 and 10, revised Supplementary Information). Those have proven the feasibility of realizing Gaussian-like functions and in-memory computing. However, fabricating fully integrated physical arrays remains prohibitively expensive for most academic laboratories, a

common challenge across the industry. As a result, investigating individual devices and validating the proposed architecture through simulations represent a critical and practical approach (*Nat. Electron.* **6**, 870-878 (2023); *Nat. Electron.* **7**, 705-713 (2024); *Nat. Commun.* **14**, 468 (2023)).

Our simulation experiments, based on realistic GMC device properties, not only offer important reference data, but also provide practical solutions for hardware system design and optimization. Through these strategies, we've determined key parameter requirements, optimized the architecture, and reduced the simulation-to-physical gap by introducing random noise. These efforts confirm G-KAN's feasibility, lay a solid theoretical and design foundation for future chip manufacturing, and mitigate risks and costs in subsequent physical implementation.

This discussion has been also added to the revised main text on page 10 and Supplementary Note 5 in the revised Supplementary Information.

Comment 2-4: The manuscript does not sufficiently address whether the scalability and stability of the GMC array can support a larger KAN for more complex intelligence tasks. A further theoretical analysis of the scalability limits of the GMC array would be valuable.

Response 2-4: We thank the reviewer for the insightful comments. It is indeed very important to take measures to verify the stability of GMCs. Experimental characterization of individual GMC devices constructed on 2D materials demonstrates

robust and reproducible Gaussian-like conductance profiles, indicating that each cell can operate reliably. This device-level stability supports the feasibility of constructing large-scale arrays. Moreover, preliminary tests on a prototype composed of two GMC units show tunable and gradually varying activation functions, indicating the potential for learning capability, as shown in Fig. R8.

Regarding the theoretical analysis of scalability limits, we agree that this is an important direction. One promising approach is to develop a modeling framework that accounts for device variability and analyzes how key performance metrics—such as accuracy or robustness—change with network scale. We decided to extend our existing simulation models by incorporating non-idealities originated from real devices to quantify performance at larger scales. In Response 2-6, we have provided an analysis based on these modeling results.

Figure R8 | Endurance performance and programmable activation function construction using parallel GMC units. **a** 1000-cycle endurance tests. **b** A parallel

circuit of two GMCs. **c** Curves of current versus gate input voltage for two parallel GMC units after applying 2, 4, and 6 pulses to individual GMCs, and the constructed activation functions (colored).

This discussion has been added to the revised main text on page 7, Fig. R8a has been included as Fig. 2f to the revised main text. Fig. R8b and c have been included as Supplementary Fig. 7.

Comment 2-5: The physical experiments presented in the manuscript focus solely on the performance of a single device. However, the manuscript does not discuss whether the proposed GMC array relies on consistency between different units and whether this could lead to additional hardware overhead. There is a lack of relevant discussion regarding the robustness of the proposed structure.

Response 2-5: Thank you for your insightful comment. To validate the feasibility of our proposed GMC-based KAN architecture and provide realistic device characteristics for further application simulation, our experiments primarily focus on evaluating the basic functionality and performance of actual GMCs. This allows the weighted coefficients of these cells to be dynamically adjusted during training to automatically compensate for effects such as current drift caused by device variations (*Nature* **577**, 641-646 (2020); *Nat. Commun.* **9**, 2385 (2018)). Such adaptability significantly reduces the reliance on device uniformity and eliminates the need for additional calibration circuits.

To assess the robustness of our proposed structure, we added additional experiments to evaluate the robustness of the KAN implementation based on GMC under device-to-device (D2D) variability. Considering that the computational error caused by D2D variability, as well as the errors induced by noise in analog devices, are both due to non-idealities in the cell output. Thus, the corresponding results and analyses are discussed in detail in our unified response to Comment 2-6. The results show that although increased device variability introduces some performance degradation, the decline is gradual and remains within an acceptable range.

In conclusion, the impact of GMCs' variability can be automatically compensated during in-situ training, without significantly affecting the performance of G-KAN. Therefore, no additional hardware overhead is required to enforce perfect consistency.

Comment 2-6: The GMC-based KAN proposed in the manuscript is based on real-world analog devices. However, any analog device inevitably introduces errors, especially in large-scale arrays, where the impact of errors becomes more significant. The manuscript lacks a discussion and analysis of these errors, which raises doubts about the feasibility of the proposed architecture.

Comment 2-6: We sincerely thank the reviewer for this insightful and constructive comment regarding the potential impact of analog device errors on the proposed GMC-based KAN architecture. As with any analog computing system, real-world implementation inevitably exists non-idealities, especially in large-scale arrays.

Against this background, a central advantage of the proposed architecture lies in

its use of programmable memristors, enabling each GMC's learnable coefficient to adaptively compensate for non-ideal effects—such as current drift induced by device variability—during the training process (*Nature* **577**, 641-646 (2020); *Nat. Commun.* **9**, 2385 (2018)). This intrinsic adaptability substantially mitigates the influence of analog device imperfections on system performance.

In addition, in our proposed architecture, the observed errors can be attributed primarily to two sources: (1) device variability, including both device-to-device and cycle-to-cycle variations; and (2) external disturbances that introduce random fluctuations. Both types of errors manifest as non-idealities in the Gaussian-like output of the GMC and can be modeled by introducing the coefficient of variation (C_V) (*Nat. Commun.* **15**(1), 129 (2024)) into the amplitude, σ_+ , and σ_- of realistic GMC devices (Fig. R9a-c). This strategy allowed us to simulate a series of Gaussian-like activation functions with computational noise (Fig. R9d), which derives from the non-idealities of realistic devices.

To evaluate the impact of errors on the performance of G-KANs, we conducted two typical task types: regression and classification tasks. In the 1D function regression task, despite the injected noise, the G-KAN maintained a high level of fitting precision, indicating minimal performance degradation after in-situ learning. To further evaluate the robustness of the G-KAN's model, the C_V values were scaled by a factor q (denoted as $q \times C$, which refers to scaling these C_V s to q times their original values) to simulate different levels of non-idealities. Note that C denotes the set of C_V , i.e., $\{C_{amp}, C_{\sigma_-}, C_{\sigma_+}\}$. When $q=1$, it indicates that only the intrinsic variability of the device is

considered, whereas $q > 1$ implies the presence of additional external disturbances. As shown in Fig. R10a, the RMSE gradually increases with the factor q from 1 to 50 on the 1D function regression task. Even under the noise levels increased up to 50 times the intrinsic case, the RMSE on this task remained below 0.4, demonstrating the robustness of the G-KAN architecture under severe variation conditions. Furthermore, we also evaluated noisy G-KAN's performance in pattern classification task. As illustrated in Fig. R10b, the learning curves under different noise presets show that the classification performance also remained robust under realistic noise conditions introduced by GMCs' variability and external disturbances. On the other hand, in response to Comment 2-4, when incorporating the intrinsic non-idealities of GMCs, the G-KAN was evaluated on a more complex classification task by enlarging the architectural scale. The results demonstrate that systematic performance was scarcely affected, as illustrated in Fig. R10c.

Figure R9 | Non-idealities of GMCs and the generated Gaussian-like curves. Fitting

statistics for 1000-cycle endurance tests. **a** I_{peak} , **b** σ_+ and **c** σ_- . **d** Visualization of the Gaussian-like functions generated by the GMCs after incorporating analog noise.

Figure R10 | Robustness evaluation of the G-KAN under device non-idealities and increasing architectural scalability. **a** 1D function regression performance under variable coefficients of variation. **b** Performance of G-KAN ([784, 64, 10], $n_c = 4$) on the pattern recognition task (MNIST) with various C_V s of device inconsistency introduced. **c** Learning curves recorded under the incorporation of GMC non-idealities as scalability increases (on FMNIST dataset).

The contents in response to Comment 2-5 and 2-6 have been added to the revised main text (page 7 and 10). Fig. R9a-c have been included as Supplementary Fig. 6, and Fig. R9d has been included as Supplementary Fig. 14a. Fig. R10 has been included as Supplementary Fig. 14b-d.

Comment 2-7: In the experiment validating the continual learning capability of the GMC-based KAN, the presented 1D symbolic regression task does not provide strong evidence. Factors such as task correlation and potential parameter redundancy may interfere with the conclusions. The study lacks comparative experiments or continual learning experiments across different 1D regression tasks to make the conclusion more

convincing.

Response 2-7: Thanks for your profound comments, which are crucial for the rigor and scientific validity of this study. We have added additional experiments in two aspects: parameter redundancy and task correlation to alleviate readers' concerns in these regards:

(1) Parameter redundancy

G-KAN has more parameters than M-MLP in the same network architecture (arXiv: 2404.19756 (2024)). Therefore, if M-MLP, even with the same or even more parameters, still fails to achieve continual learning like KANs, this would help rule out the concern about parameter redundancy. As depicted in Fig. R11a, since the G-KAN was configured with 100 basis functions, the M-MLP initially preset a size of [1, 100, 1] to match the parameter count of both models. However, as clearly observed in Fig. R11b, the M-MLP failed to achieve even basic fitting, let alone continual learning. Additionally, after expanding the depth of the M-MLP sufficiently, the well-formed fitting Gaussian functions can be finally observed. Regrettably, each time the model learned a new Gaussian peak, it forgot all previous peaks, demonstrating that the M-MLP completely lacks the capability for continual learning in 1D function regression scenarios (Fig. R11c). These results further solidify the recognition that KANs, with their local activation capabilities, inherently possess continual learning as a functional attribute, in contrast to MLPs, which rely on global activation.

Figure R11 | The continual learning capability validations of the G-KAN and M-MLP on 1D function regression. **a** Fitting results of G-KAN ([1, 1], $n_c = 100$). **b** Fitting results of M-MLP [1, 100, 1], showing no successful fitting. **c** Fitting results of M-MLP [1, 50, 50, 50, 200, 1], exhibiting very severe catastrophic forgetting.

(2) Task correlation

In previous experiments, each function period in a dataset was obtained via translation, and all periods consisted entirely of Gaussian functions. To mitigate

potential concerns regarding task correlation, we ensured that the data within different periods of the dataset did not completely overlap, and also introduced datasets generated by other functions. Based on this approach, function regression performed on this basis can validate whether the continual learning capability of G-KANs is independent of task correlation. This scenario encompasses three cases: non-Gaussian periodic, non-periodic, and piecewise functions.

Specifically, Fig. R12a presents a periodic function, but since each half wave is treated as a separate subset, the training samples in each period do not completely overlap. In contrast, Fig. R12b presents a non-periodic function, under which G-KAN continues to exhibit strong continual learning capability. Fig. R12c represents a more challenging case, in which the G-KAN is trained on a piecewise function with intentionally introduced discontinuities between segments to minimize task correlation. As illustrated in this subplot, even when presented with unrelated sequential data, the G-KAN still demonstrates effective continual learning performance. Apart from a slight impact near the discontinuous boundaries, no significant catastrophic forgetting can be observed throughout the whole process. From the three subplots in Fig. R12, it is clearly observed that whether the dataset consist of periodic, non-periodic, or piecewise functions, G-KANs can perfectly fit the ideal function curves. This set of experiments demonstrated that, the capability of G-KANs to achieve continual learning is not a result of task correlation.

Based on the experimental results mentioned above, the continual learning capability of G-KANs is independent of parameter redundancy and task correlation.

Figure R12 | Fitting performance of the G-KAN on non-repetitive function datasets.

Models are trained by a single non-Gaussian periodic function (a), a non-periodic function (b), and a piecewise discontinuous function (c).

This discussion has been added to the revised main text (page 10 and 11) and Supplementary Note 3. Fig. R11 has been included as Supplementary Fig. 15, and Fig. R12 has been included as Supplementary Fig. 11 to the revised Supplementary Information.

Comment 2-8: In the comparison between KAN and MLP, the parameter counts corresponding to the same network size (784-64-10) for KAN and MLP are not comparable (KAN has more parameters). The specific details of the network configurations are not presented to demonstrate the fairness of the comparison. To ensure a fair comparison, more details about the network structures and parameter settings should be provided.

Response 2-8: Thank you for your insightful comment. Our original intention was to ensure fairness in the comparison by aligning the network structures, meaning that KAN have the same number of layers and neurons as M-MLP. We also acknowledge that under this configuration, G-KAN have more parameters due to its architectural design.

To ensure a more comprehensive comparison, we conducted additional experiments that allowed us to evaluate G-KAN and M-MLP under an identical parameter budget. Specifically, we redesigned the network architectures as G-KAN [784, 100, 10] ($n_c = 4$) and M-MLP [784, 500, 10], ensuring both models have the same count of parameters. As shown in Fig. R4, G-KAN achieves classification accuracy comparable to that of M-MLP.

However, we would like to emphasize that beyond pattern recognition tasks, G-KAN shows clear advantages over M-MLP in more scenarios, such as 1D function regression, partial differential equation (PDE) solving, and time-series forecasting. These tasks demand that the computing system generate continuous and accurate numerical outputs with a high dynamic range, such as floating-point (FP) numbers (*Nat. Electron.* **8**, 276–287 (2025)), where G-KAN’s architectural design offers tangible benefits. In addition, the corresponding research content can be found in detail in response to Comment 2-10.

It is also worth noting that conventional computing hardware is not yet optimized for the KAN architecture. Therefore, the main contribution of our work lies in proposing a hardware-efficient implementation tailored specifically for KAN, which we believe will help unlock its full potential in future applications.

Figure R4 | The learning performance of G-KAN ([784, 100, 10], $n_c = 4$) is comparable to that of M-MLP [784, 500, 10] (equal training parameters) on image classification task. (Copied from response to Comment 1-2)

Comment 2-9: In the MNIST handwritten digit classification task, as shown in the training curves (Fig. 4c and 4e), it appears that neither the KAN nor the MLP training process has fully converged. Therefore, such results cannot rigorously demonstrate the performance differences between the two methods.

Response 2-9: Thanks for the suggestion. We have increased the training epochs in the MNIST handwritten digit classification task from 10 to 50. Under the same number of parameters in M-MLP as in G-KAN, the performance of the two models became comparable. We discussed it in detail in response to Comment 2-8. To further explore the advantageous applications of G-KANs, we compared the performance of G-KANs and M-MLPs in additional scenarios. Specifically, we successfully validated the advantages of G-KANs on PDE and time-series tasks. Detailed information can be found in the response to Comment 2-10.

We have added the discussion in response to Comment 2-8 and 2-9 to the revised main text (on page 12 and 13). Fig. R4 has been included as Fig. 4c in the revised main text.

Comment 2-10: The symbol regression and MNIST image classification tasks presented in the manuscript have shown the basic intelligence computing capacity for the GMC-based KAN. However, these tasks could also be well-handled by conventional MLP or CNN networks. To proof the advances of the proposed physical KAN, extra tests on more complex tasks e.g., CIFAR or ImageNet tasks will be

appreciated.

Response 2-10: We thank the reviewer for the valuable suggestions. We evaluated both M-MLP and G-KAN on the CIFAR-10 dataset (*Nat. Commun.* **16**, 702 (2025)) and found that neither model performed satisfactorily (Fig. R13a). This situation has already been demonstrated in previous literature as well (*arXiv:2407.16674* (2024)), and this problem can be addressed by extending these foundational frameworks—for instance, by incorporating residual blocks to handle such complex image datasets (*arXiv:2410.05500* (2024)). However, this lies beyond the scope of this work, and we plan to explore it more thoroughly in future studies. To better demonstrate the advancements of our proposed GMC-based KAN, we designed two more challenging computational tasks: partial differential equation (PDE) solving and time-series forecasting.

On the one hand, for the PDE task, we considered Poisson equation with zero Dirichlet boundary conditions data. For a space detailed as $\Omega = [-1, 1]^2$:

$$\begin{cases} u_{xx} + u_{yy} = f & \text{in } \Omega, \\ u = 0, & \text{on } \partial\Omega \end{cases} \quad (\text{R3})$$

Considering the data $f(x, y) = -2\pi^2 \sin(\pi x) \sin(\pi y)$ for which the exact solution is $u = \sin(\pi x) \sin(\pi y)$. Using the framework of physics-informed neural networks (PINNs) to solve this PDE, with the loss function given by:

$$loss = \alpha loss_i + loss_b = \alpha \frac{1}{n_i} \sum_{i=1}^{n_i} |u_{xx}(z_i) + u_{yy}(z_i) - f(z_i)|^2 + \frac{1}{n_b} \sum_{b=1}^{n_b} u^2 \quad (\text{R4})$$

where $loss_i$ is the interior loss, discretized and evaluated by a uniform sampling of n_i points $z_i = (x_i, y_i)$ inside the domain, and similarly $loss_b$ is the boundary loss evaluated at n_b uniformly sampled points on the boundary. The parameter α balances the two loss components. We compared G-KAN with two different M-MLP architectures using the same hyperparameters ($n_i = 51$, $n_b = 51$, $\alpha = 0.01$). As shown in Fig. R13b, G-KAN achieves a significantly lower L2 norm error while using a smaller network size and fewer parameters. Specifically, G-KAN attains an L2 norm error of 0.0121, which is much lower than that of M-MLP, which is 1.0121 and 0.2215, respectively.

On the other hand, for the time-series forecasting task, we used a chaotic system as benchmark test. It is inherently very challenging due to the positive Lyapunov exponent (*Transp. Res. Rec.* **1897**, 9-17 (2004)), which leads to exponential growth of separation of close trajectories so that even small errors in prediction can quickly lead to divergence of the prediction from the ground truth. We tested the system using the Mackey-Glass time series (*Nat. Electron.* **2**, 480-487 (2019); *Science*, **197**, 287-289 (1977)):

$$\frac{dx}{dt} = \beta \frac{x(t-\tau)}{1+(x(t-\tau))^n} - \gamma x(t) \quad (\text{R5})$$

Herein, the parameters were set to $\beta = 0.2$, $\gamma = 0.1$, $\tau = 18$, and $n = 10$, respectively. G-KAN and M-MLP were trained to predict the next time step based on the previous 20 time steps. Fig. R13c presents the results, G-KAN's predictions demonstrate strong agreement with the ground truth, with a root mean square error (RMSE) of 0.0691. In contrast, even with increased network depth and width, the RMSE of M-MLP

consistently remained above 0.6, indicating its difficulty in accurately forecasting chaotic systems.

Figure R13 | Performance evaluation on other typical complex tasks. **a** Performance comparison between G-KAN and M-MLP on the CIFAR-10 dataset. **b** PDE example: the squared L2 loss between the predicted solution and the ground truth. G-KAN achieves much lower error compared to M-MLP. **c** Time-series forecasting example: training (blue) and forecasting (pink) results of G-KAN and M-MLP.

This discussion has been added to the revised main text (page 13), and Supplementary Note 8. Fig. R13 has been included as Fig. 4 in revised main text, and Supplementary Fig. 18.

Response to Reviewer 3's comments

General comment: The manuscript proposes a Gaussian-like memory cell (GMC), consisting of a Gaussian transistor and a memristor, as the basis function for Kolmogorov-Arnold Networks (KAN), an emerging neural network algorithm. While exploring efficient hardware implementations of KAN is an important topic, this manuscript only scratches the surface, relying on pure simulation data and very limited experimental results from a single cell. Further analysis and more comprehensive studies are necessary for this manuscript to be considered for publication.

Response: We sincerely appreciate your insightful comments and constructive feedback on our manuscript. In response to your concerns regarding the limited experimental validation and simulation depth, we have expanded both the experimental and simulation components of this work. Specifically, we have conducted additional device-level measurements on multiple fabricated GMCs to evaluate performance consistency and reliability, performed comprehensive simulations to demonstrate the feasibility of the proposed GMC-based KAN architecture in real-world applications, and added stability and noise analysis to validate the robustness of the design under practical operating conditions. We believe these improvements will address your concerns and provide a more complete and rigorous evaluation of the proposed approach.

Comment 3-1: The core contribution of this work is the development of the GMC. However, the manuscript only provides minimal experimental data, such as the transfer curve of the memory cell in Fig. 2. Additional device characterization, such as endurance testing etc., is recommended to demonstrate the reliability and learning capability of the proposed memory cell. In other words, what are the full suite of device properties that are required to run the KAN algorithm?

Response 3-1: We are grateful to the reviewer for the insightful comment, which is crucial for us to elaborate on the electrical performance of the GMC and how to utilize GMCs to construct the core of KANs - the variable activation function.

To implement the KAN's algorithm on hardware based on GMC, each cell must satisfy several key requirements, including realization of Gaussian-like curve, multistate storage capability, stable retention characteristics, and high endurance. The weighted linear superposition of Gaussian-like curves is critical to KAN's implementation as it can approximate any activation function. As shown in Fig. 2d in the original manuscript, the GMC can have satisfied Gaussian-like curves with tunable weighting coefficients. Furthermore, the retention characteristics and endurance are essential for KAN's hardware implementation to ensure hardware stability and reliability. In the revised manuscript, we have provided tests about the GMC's retention in 100 seconds and endurance in 1000 cycles (Fig. R14). Through fitting and statistical analysis of the endurance test, we found that the coefficient of variation (*Nat. Commun.* **15**(1), 129 (2024)) (C_v) for the maximum current, μ , and σ_{\pm} of the GMC are all within 0.075. To demonstrate GMCs' feasibility in a whole integrated architecture, we have

incorporated random noise testing in simulation, and the results shown that the performance of G-KAN remained stable without significant degradation within this C_V range.

Moreover, to intuitively demonstrate the learning ability of the GMC array, we fabricated a 1×2 device array to implement activation functions with continuously tunable characteristics. This array configuration serves as the minimal complete unit required for approximating arbitrary activation functions in G-KAN, providing the essential functional building blocks for constructing any nonlinear mapping. By applying 2, 4, and 6 pulses to each GMC, we modulated their weighted states, enabling precise control over the shape of the activation functions. The evolution of the output functions is illustrated in Fig. R15. The results verified the feasibility of approximating any activation function through the linear combination of multiple GMCs.

Figure R14 | Endurance tests. a Results of the endurance test in 1000 cycles. Fitting statistics for endurance tests in 1000 cycles: **b** I_{peak} , **c** σ_+ , and **d** σ_- .

Figure R15 | 1×2 GMC array achieving a learnable activation function. **a** A parallel circuit of two GMCs. **b** Curves of current versus gate input voltage for two parallel GMC units after applying 2, 4, and 6 pulses to individual GMCs (grey), and the constructed activation functions (colored).

We added this discussion to the revised main text (page 7). Fig. R14a has been included as Fig. 2f in the revised main text, and Fig. R14b-d have been included as Supplementary Fig. 6. Fig. R15 has been included as Supplementary Fig. 7. In addition, the original Fig. 2e has been relocated to Supplementary Fig. 4, while the newly added cycle-to-cycle characteristics have been presented in the revised Fig. 2.

Comment 3-2: The circuit design presented in Fig. 3 is more of a functional block representation than a detailed circuit implementation. To adequately demonstrate the circuit-level design, the manuscript should include explicit circuit diagrams for the key blocks rather than just describing their functions. More importantly, a circuit capable of running the KAN should be experimentally built and tested.

Response 3-2: Thank you for the suggestion, which is helpful in improving this work.

In Fig. 3a-d of the manuscript, the circuit of the G-KAN was presented in a modular form. This type of illustration may obscure some of the more detailed designs. To provide a clearer illustration of the circuit details, a complete array design is presented in Fig. R18. The following content gives detailed explanation about the working principle of the circuit from input to output in sequence for the entire workflow.

Firstly, the translation module provides Gaussian-like basis functions with evenly spaced center positions. The principle relies on translating the input signal to induce a corresponding translation in the function. Based on this mechanism, the sequence of translation terms can be derived through function construction and translation properties, as detailed below.

Given that the device transfer characteristic is described by the function

$f(V_g) = \exp\left(-\frac{(V_g - \mu)^2}{\sigma^2}\right)$, and assuming the target basis function is expressed as

$g(u) = \exp\left(-\frac{(u - \mu)^2}{(\sigma/k)^2}\right)$. Herein, $k=L/4$ (L is the grid size, and the value '4'

originates from the transfer characteristic of the GMC), since the GMC intrinsically corresponds to a grid size of 4, ensuring that the width of each grid cell exactly matches the parameter σ . Therefore, equating $f(V_g) = g(u)$ yields:

$$V_g = ku - (k-1)\mu \quad (\text{R6})$$

Define $h(x) = \exp\left(-\frac{k^2x^2}{\sigma^2}\right)$. It is evident that $h(x + \Delta x) = \exp\left(-\frac{k^2(x + \Delta x)^2}{\sigma^2}\right)$. Let

$g(u) = h(x + \Delta x)$, then the result follows that $u = x + \Delta x + \mu$. Therefore,

$V_g = k(x + \Delta x) + \mu$. Since the translated centers of $h(x)$ are evenly distributed across L

points within the interval $[-1,1]$, the set of Δx can be calculated as:

$$\Delta x \in \{1, 1-\delta, 1-2\delta, \dots, -1+2\delta, -1+\delta, -1\} \quad (\delta = \frac{2}{L-1}) \quad (R7)$$

In summary, the voltage sequence applied to the gates of different GMCs, according to the input signal (x) , is given by that:

$$V_g \in \{k(x+1) + \mu, k(x+1-\delta) + \mu, \dots, k(x-1+\delta) + \mu, k(x-1) + \mu\} \quad (R8)$$

Moreover, in Fig. R16, each original input signal (x) enters along the red path and is distributed equipotentially across parallel GMC columns. For the independent memristor column, the non-negative input signal, i.e., $\text{ReLU}(x)$, is applied through a multiplexer (MUX). In addition, each one-transistor-one-resistive-memory (1T1R) unit contains a pair of amplification gains of $-A$ and A , respectively. Herein,

$A = \frac{I_{\max}^{\text{GMC}} - I_{\min}^{\text{GMC}}}{I_{\max}^{\text{memristor}} - I_{\min}^{\text{memristor}}}$, which serves as a normalization factor. The alternating placement of gains is designed to match the polarity of the output lines in the GMC array.

For the signal routing, equipotential paths including VBL, VWL, VSL, INPUT, and GND, are employed with configurable interface switches to specify the desired connection combinations. For example, during the forward propagation, the potential on the VWL is set to V_d , supplying power through the drain of GMC. On the other hand, during backpropagation, the potentials coordinate with the selected cells and the write voltage pulses to determine the target cell locations and the polarity of the pulses (SET or RESET). In array techniques, signal routing can be configured utilizing field-programmable gate arrays (FPGAs) in conjunction with peripheral circuits (*Nature* 2020, 577(7792), 641-646).

Figure R16 | The complete circuit design of a G-KAN.

Based on the above design, a modular system comprising the translation module, $b(x)$ -row module, and the GMC array was constructed on the Simulink platform (Fig. R17a). Additionally, Fig. R17b presents the oscilloscope image of the linear input signal (x) from -1 V to 1 V, while the output waveform of the array is shown in Fig. R17c.

Figure R17 | Modular systematic architecture and testing results of the G-KAN (with a size of [1,1], and the $n_c = 100$) in Simulink. **a** Diagram of the architecture. **b** Oscilloscope image of the input signal $x(t)$ from -1 to 1 (V). **c** Oscilloscope waveform of the output signal $G\text{-KAN}(x(t))$.

We added this discussion to the revised main text (page 8 and 9), and Supplementary Note 1. Fig. R16 and Fig. R17 have been included as Supplementary Fig. 8 and 10 in the revised Supplementary Information, respectively.

Comment 3-3: Are the results shown in Fig. 3e from experimental setups or pure simulations? If they are based on experimental setups, a photo of the memory cell array should be included to support the findings. The manuscript should provide clear details on the simulation setup and parameters if they are from simulations.

Response 3-3: Thank you for your attention to this information. The data in Fig. 3e was sourced from simulations, which we have validated both in Python and using the Simulink platform based on very high-speed hardware description language (VHDL). Notably, the simulations were entirely based on the polymorphic results obtained from the actual measurements of GMCs and CIPS memristors. There are the additional details regarding the execution of the simulations below.

In the 1D function regression task, since the grid size is expanded to 100 (i.e., expanded 25 times). Accordingly, in the translation module, $k = 100/4 = 25$. Specifically, the training dataset consists of five segments of Gaussian peaks ($f_i(x) = \exp[-300(x - c_i)^2]$ ($i = 1, 2, 3, 4, 5$), where the $f_i(x)$ and c_i represent the function and the center of the i -th Gaussian peak), with each segment containing 1,000 data points. The G-KAN was trained for 1,000 iterations on the data from each segment of the Gaussian peaks using the L1 loss function and AdamW optimizer with a learning rate of 0.02 and weight decay of 0.0001. During this process, all learnable coefficients (c) were encoded by the differential pairs of GMCs, while the weights (w_b) were encoded by the differential pairs of CIPS memristors. Furthermore, in Simulink, the translation module receives a scanning signal ranging from -1 V to 1 V generated by a

signal generator and distributes the processed signals to each row of GMC pairs. The device responses are accumulated along the differential columns and the output can be read out via an oscilloscope.

Comment 3-4: The claim regarding “continual learning resilience against catastrophic forgetting” in Fig. 3 appears to be an inherent benefit of the KAN rather than the GMC. To substantiate this claim, a comparative analysis should be conducted by evaluating the performance of the same KAN structure using the GMC versus other basis functions. Otherwise, this claim only reiterates the advantages of KAN without demonstrating the specific contributions of the GMC.

Response 3-4: Thank you for your valuable comments. Indeed, the catastrophic forgetting resistance is inherent advantage of KANs. However, our work main emphases are on how GMCs enable the hardware efficient implementation of KANs at the analog level, while simultaneously enabling parallel computing in memory (CIM).

Moreover, we have included experimental comparisons between B-splines and GMC-based implementations in a 1D function regression task to evaluate their performance. Both scenarios were evaluated under identical conditions with the same loss function, optimizer, and hyperparameters. The performance of the G-KAN was almost identical to that under the B-spline functions (Fig. R18). It is worth noting that, at the physical level, GMCs can serve as basis functions that are more straightforward and hardware-friendly to implement on analog platforms. In contrast, B-splines rely on the recursive Cox-De Boor formulation, which requires considerable hardware

resources for implementation (see Table R1). Moreover, G-KAN is specifically designed based on a CIM architecture, allowing KANs to operate efficiently in both the analog signal domain and memory-centric computation frameworks. Specifically, the comparison of time consumption between using GMCs or B-splines is illustrated in Fig. R7.

Figure R18 | The performance comparison of KANs with different basis functions. **a** The performance of a KAN when using B-splines. **b** The performance of a G-KAN based on GMCs.

Figure R7 | Statistics on the time consumption of basis functions based on GMCs or B-splines in KANs. (Copied from the response to Comment 2-1)

Table R1 | Comparison of different ways to implement spline functions. (Copied from the response to Comment 2-1)

Ref.	Mode	function	Amp. adjustability	Memory	Num. of devices	Area (mm ²)	Power (μW)
Nakashima et al.	digital	B-Spline	No	No	LUTs* (96)	-	-
Yu et al.	digital	B-Spline	No	No	LUTs* (96)	-	-
Lozano Duarte et al.	digital	B-Spline	No	No	-	9.111	266.735
Lozano Duarte et al.	analog	B-Spline	No	No	42	0.073	238.5
This work	analog	Gaussian	Yes	Yes	2	2.2×10⁻⁴	~10⁻³

* The transistor count estimation for each 4-input LUT (4-LUT) is based on the architecture of Xilinx Virtex-4 FPGAs, where each 4-LUT can be configured as a 16-bit distributed RAM. Assuming each SRAM bit cell comprises six transistors, the total transistor count per 4-LUT can be approximated by multiplying the number of SRAM cells (16) by six, resulting in approximately 96 transistors per LUT.

We added this discussion to the revised main text (page 10). Fig. R7 and Fig. R18 have been included together as Supplementary Fig. 13.

Comment 3-5: Similarly, beyond simply demonstrating that G-KAN can perform classification tasks like KAN, the manuscript should include studies on the trade-offs introduced using the GMC. Specifically, the impact of computing noise from the analog device on the performance of G-KAN should be evaluated, as well as its generalization capability to other tasks when using only a Gaussian-like function as the basis function.

Response 3-5: We appreciate the reviewer's constructive comments. In practical applications, computational noise is inevitable due to external interference and non-ideal variability inherent in analog devices, which can affect hardware performance.

Hence, we introduced perturbations at the output of the GMC to simulate the effects of external interference and device-level variability, with the aim of quantitatively assessing the influence of computational noise on the G-KAN performance. As computational noise from external interference generally follows a Gaussian distribution and can be compounded with device-level variability, the latter was used as a representative factor to account for both effects in our analysis. In detail, we measured the coefficients of variation (C_V) of GMCs, as shown in Fig. R9 a-c. Based on this measurement, the Gaussian-like functions generated by GMCs exhibited non-ideal states, as illustrated in Fig. R9d. After introducing computational noise, the G-KAN still exhibited relatively strong robustness and tolerance in the 1D function regression task. Note that C denotes the set of C_V , i.e., $\{C_{amp}, C_{\sigma_-}, C_{\sigma_+}\}$, and $q \times C$ refers

to scaling these C_V s to q times their original values. As shown in Fig. R10a, the RMSE gradually increases with the factor q from 1 to 50 on the 1D function regression task. Even under the noise levels increased up to 50 times the intrinsic case, the RMSE on this task remained below 0.4, demonstrating the robustness of the G-KAN architecture under severe variation conditions. The robustness of the G-KAN was also evaluated in pattern recognition tasks, addressing the same concern raised by Reviewer #2, as illustrated in Fig. R10b and c referenced earlier. The analysis shows that under noise conditions induced by C_V within 10C, the accuracy degradation of the G-KAN remains relatively minor.

The above supplementary experiments are provided to further support the effectiveness and advantages of G-KAN.

Figure R9 | The output of GMC is not an ideal Gaussian-like curve. Fitting statistics for 1000-cycle endurance tests. **a** I_{peak} , **b** σ_+ and **c** σ_- . **d** Visualization of the Gaussian-like functions generated by the GMC after incorporating analog noise. (Copied from

the response to Comment 2-6)

Figure R10 | Robustness evaluation of the G-KAN under device non-idealities and increasing architectural scalability. **a** 1D function regression performance under variable coefficients of variation. **b** Performance of G-KAN ([784, 64, 10], $n_c = 4$) on the pattern recognition task (MNIST) with various C_V s of device inconsistency introduced. **c** Learning curves recorded under the incorporation of GMCs' non-idealities as scalability increases (on FMNIST dataset). (Copied from the response to Comment 2-6)

This discussion has been added to the revised main text (page 7 and 10). Fig. R9a-c has been included as Supplementary Fig. 6, and Fig. R9d has been included as Supplementary Fig. 14a. Fig. R10 has been included as Supplementary Fig. 14b-d.

Reviewer #1 (Remarks to the Author):

I'm glad to see that authors have addressed all my concerns.

Response: We sincerely appreciate the reviewer's positive feedback and pleased that we were able to address your concerns. We also truly appreciate the reviewer's constructive comments and suggestions throughout the previous review process, which have been highly valuable in improving the clarity, rigor, and overall quality of our work.

Reviewer #2 (Remarks to the Author):

The revised manuscript provides more comprehensive details regarding the implementation of the proposed G-KAN and addresses several of my previous concerns. However, there are still some important issues that need to be addressed.

Response: We sincerely thank the reviewers for these constructive comments on our revised manuscript. We are delighted that the previous improvements have already received positive responses. Reviewer's comments have been valuable in helping us further clarify the key contributions of this work and strengthen its overall presentation. We carried out a more detailed revision process, carefully addressing each suggestion point by point. We conducted additional experiments to enhance the rigor of our work and ensure that the narrative can more clearly convey the significance of our findings. We hope these improvements further demonstrate the potential of the GMC-based KAN framework.

1. While the response includes a comparison between G-KAN and M-MLP to demonstrate performance advantages, these benefits appear to stem largely from the inherent characteristics of the KAN architecture itself. The revised manuscript still provides limited discussion on the specific advantages of the GMC-based implementation. More thorough discussion and comparative visual evidence are required to underscore the contribution of the proposed GMC implementation.

Response 2-1: We sincerely appreciate the insightful comments provided by the reviewer, which have offered a valuable opportunity to further clarify and enhance the significance of our research. In the revised manuscript, we have included a comparative analysis of typical hardware implementation approaches to highlight the significant and adaptability advantages of GMCs in executing KAN-type networks.

As Gaussian kernels constitute the most underlying computational units in KANs, the key factor in determining whether KANs can be effectively hardware-implemented lies in how effectively the Gaussian kernels' computation and training can be realized through devices or circuits. However, existing hardware architectures (von Neumann and near-memory architectures) face significant bottlenecks, both in implementing the Gaussian kernels and in training, which ultimately hinder the full potential of KANs. The proposed GMCs are specifically introduced to overcome the challenges that current devices and architectures cannot address (Fig. R1). To fully highlight the significance of GMC-based implementation for hardware KANs, the following section will present real-world cases to demonstrate why conventional hardware architectures are unsuitable for implementing KANs, thus contrasting the advantages of G-KANs.

(1) Dilemmas of von Neumann architecture (corresponding to Dilemma 1-3 in Fig. R1b)

In the von Neumann architecture, the physical separation of the processor and memory requires continuous data and instruction transfer between them during execution. Since memory bandwidth is much lower than the processing throughput, processing cores often remain underutilized while waiting for data.¹ Moreover, accessing memory incurs significant time and energy costs, with much of the energy spent on data transfer rather than computing (**Dilemma 1 and 2**).^{1,2} Additionally, the graphics processing unit (GPU) is a representative example of optimized computing design. GPUs have become prevalent for their remarkable acceleration in general matrix multiplication (GEMM)³. In neural network computations, GPUs can perform linear transformations with high computational density by exploiting highly regular operations and contiguous memory access⁴, while the central processing unit (CPU) only needs to initiate a single scheduling process, minimizing offloading overhead.⁵ However, these capabilities are not fully leveraged in KANs' computation. Taking Gaussian kernels as the basis functions, the inference process involves sequential operations, which contain squaring ($(x-a)^2$), scaling multiplication ($-\frac{1}{\sigma} \cdot (x-a)^2$), nonlinear exponential transformation ($\exp(\cdot)$), weighted multiplication ($c_i \cdot \exp(\cdot)$), and accumulation (\sum_i). These operations cannot be directly handled by GEMM, requiring frequent kernel launches per feedforward processing (each scheduled by the CPU), significantly increasing CPU-GPU offloading overhead (**Dilemma 3**). Worse yet, KAN's complex computational components consume more hardware resources and

power compared to MLPs, even with the same number of parameters.^{2,6}

(2) Dilemmas of near-memory implementations (corresponding to Dilemma 4-6 in Fig. R1b)

The evidence above confirms the incompatibility of KANs with the von Neumann architecture. Therefore, it is crucial to implement KANs on hardware platforms with higher computational and memory density. In this context, researchers have attempted to alleviate the dilemmas of von Neumann architecture through near-memory computing⁷. For KAN-related applications, typical examples include lookup table (LUT)⁸ and piecewise linear (PWL)⁹ circuit designs. Although LUT-based circuits can directly output Gaussian kernels at the hardware level, this strategy inevitably introduces significant computational complexity and storage address overhead. For instance, implementing a single Gaussian kernel requires 452 LUTs.⁸ When scaled to a network level, for highly densely integrated architectures like KANs, the resulting operational costs become unbearable (**Dilemma 4**).

Moreover, a recent study have employed CMOS integration technology based on PWL to directly approximate the activation functions (connections) in KANs.⁹ However, this strategy has several inherent bottlenecks: first, the PWL function requires predefined breakpoint (X_n), slope (M_n) and intercept (Y_n) of each segment ($X_{n-1} \leq X < X_n$), which must be stored in peripheral memories; second, the hardware architecture must integrate massive comparators (COMPs) and sensing circuits to determine which segment the input signal belongs to, then retrieve the corresponding slope and intercept from the buffers and pass these parameters to the MAC circuits to

complete the linear operation ($F(X) = M_n * X + Y_n$). Therefore, even for simple and few target operations, this approach still suffers from heavy hardware resource overhead (**Dilemma 5**).

Overall evaluation of these two strategies reveals that they not only require significant hardware resources such as multipliers (MPLs), adders (ADDs), and COMPs to execute Gaussian kernel computations, but also, due to their hardwired electrical architecture, are physically incapable of implementing parameter update functionality (**Dilemma 6**). Therefore, although both aim to alleviate the von Neumann bottleneck, LUT-based and PWL-based implementations ultimately suffer from a fundamental flaw due to their inability to endow the neural network with trainable characteristics.

In the context given above, the limitations of existing CMOS-based implementations have led to the exploration of dedicated KAN-type computing-in-memory (CIM) architectures. Clearly, memristors alone cannot achieve in-memory computation for nonlinear functions such as Gaussian kernels. The following content will specifically elaborate on how the GMCs and the array design proposed in this work effectively address the aforementioned challenges.

(3) Solutions relying on GMC-based implementation (corresponding to the Solution 1-5 in Fig. R1b).

The transfer characteristics of GMCs can directly generate stable and parameterized Gaussian kernels, naturally representing the mathematical behavior of Gaussian kernels in the analog domain. In addition, by adjusting the conductance of

memristors through write voltages, the peak height of the Gaussian kernel can be flexibly controlled, enabling electrical programming of the learnable coefficients. At the architectural level, the one-transistor-one-memristor structure of GMCs facilitates easy integration into crossbar arrays.¹⁰ According to the KCL, the output currents of all GMCs in the same column can be physically accumulated on the source line, directly performing the linear combination of Gaussian kernels in a KAN. Therefore, the GMC-based implementation can complete the full computation from input to output within the analog domain (**Solution 1**).

Furthermore, the Dilemma 2 in conventional CMOS-based implementations arises from the physical separation between processor and memory.¹ The GMC-based implementation fundamentally reshapes this paradigm through the CIM approach. The core mechanism lies in synchronizing computation and data storage at the same physical location, thus completely avoiding the frequent data transfers between memory and processing units. As a result, the conflict between memory bandwidth and processing throughput is eliminated, with energy consumption focused on computation (**Solution 2**). Meanwhile, the in-situ CIM working mode also entirely avoids the offloading overhead inherent in von Neumann architecture (**Solution 3**).

To address the hardware catastrophe scaling problem (Dilemma 4) faced by CMOS-based implementations—an inherent bottleneck where the growth in the number of basis functions triggers a nonlinear explosion in hardware resource consumption—the GMCs proposed in this study offers a fundamental solution through a linear hardware scaling paradigm. Specifically, under the differential-pair encoding

rule, each Gaussian kernel requires only a fixed pair of GMCs. This ensures that the hardware scale expands strictly linearly with the number of Gaussian kernels, which is equivalent to the network scale of a KAN. As a result, this approach fundamentally prevents uncontrollable resource consumption at the hardware level, ensuring that the hardware scale of a G-KAN remains predictable and manageable throughout (**Solution 4**).

Of equal significance, the GMC-based paradigm circumvents the complex and segmented operations that characterize near-memory computing implementations (Dilemma 5 and 6). Within the GMC arrays, an input vector is loaded in parallel across all rows, enabling each cell to perform in-situ Gaussian-like mapping and MAC operations simultaneously. By leveraging the native physics of the crossbar array and KCL, the complete matrix transformation is executed in one step, thereby obviating the need for piecewise function evaluation, parameter retrieval from memory, and data shuttling. This one-shot process fundamentally eliminates the timing and control overhead associated with multi-step operations in CMOS-based implementations, achieving a drastic reduction in computational complexity (**Solution 5**).

The primary dilemmas confronting the aforementioned CMOS-based implementations and their corresponding solutions enabled by the GMC-based implementations are graphically summarized in **Fig. R1**. Furthermore, a detailed comparison of the key attributes discussed for the reference objects is provided in **Table R1**.

Fig. R1 | Overview of KAN's hardware implementation strategies. **a** Reference object and benchmark metrics for the hardware implementation of KAN. **b** The bottlenecks encountered by the von Neumann and near-memory architectures in processing the computations of KAN, while the GMC-based implementation provides a reliable solution at the in-memory computing level. **c** The GMC ingeniously combines the Gaussian kernel provided by AAT with the parameterization capability of memristors, integrating the necessary physical foundation for the in-memory computing required by KAN.

Furthermore, in order to more systematically highlight the advantages of GMC in implementing KAN and the basis functions (i.e., Gaussian kernels) for each activation

connection, we compare different implementation paths with their reference objects and benchmark metrics. **Table R1** present a complete summary across several comparative dimensions as below.

Table R1 Comparison for implementing KAN across different benchmarks.

Reference objects/ benchmarks	Von Neumann Architecture	Near-Memory Architecture	GMC-based Crossbar Array
Parameterized Gaussian kernel implementation	 1. hardware dependance: DSPs/GPUs/FFs/RAMs/LUTs^{6,11-13}; 2. Muti-step pointwise operators (squaring → multiplication → exp() → multiplication); 3. memory-bound;^{14,15} 4. frequent memory accessing¹⁶ 	 1. memory-bound; 2. Separation of parameterization and kernel computation; 3. Non-adjustable functions (hardwired); 4. Reliance on a large number of computational modules^{8,9} 	physically in-device generation
	if one-shot?	No	No
Computing in memory	NOT	partial	complete
	if one-shot?	No	No
Pipeline	 1. Data loading and scheduling initialization (CPU side); 2. Multi-step chain of pointwise operators; 3. Weighted accumulation; 	(LUT-based)⁸ ADC → multi-bit input (address) → multi-bit output → DAC; (PWL-based)⁹ binary encoding → breakpoint detection → accessing	analog MAC implemented on the crossbar array

	4. Memory copy 	slope/intercept memory→ parameter fetching →digital MAC→ DAC 	if one-shot?	No	No	Yes
Computational Parallelism	(partially parallel on GPUs) 1. memory-bound; ¹⁷ 2. high overhead of data movement; ¹⁸ 3. significant CPU-GPU offloading latency and energy consumption; ¹⁹ 4. high overhead of scheduling and synchronization ^{20,21} 	(partially parallel)⁹ 1. memory-bound; 2. massive computational modules; 3. intricate control flow (PWL-based scene); 4. redundancy in address (LUT-based scene) or parameter (PWL-based scene) access 	(completely parallel) 1. access-free memory; 2. offloading-free; 3. one-shot parallel VMM and MAC process; 4. simple control mechanism (voltage-pulse direct drive) Hardware overhead change	1. memory and computation demand grows superlinearly with the number of parameters; 2. exponentially increasing memory copy latency; [*] 3. memory bandwidth bottleneck amplifies with increasing computational scale ²² 	1. memory and computation demand grows superlinearly with the number of parameters 	1. device count demand increases linearly with computational scale (one basis function is implemented by a pair of GMCs) In-situ training	--	NO (hardwired)	YES

			--	--	---	---

* <https://developer.nvidia.com/gdrcopy>

We have included the response in Supplementary Note 1 of the latest revision of the manuscript. Additionally, Fig. R1 is presented as Fig. 2 in the latest main text, while Table R1 is shown as Supplementary Table 1 in the latest Supplementary Information.

2. The experiment involving two GMC units in the revised manuscript only demonstrates the functional feasibility. However, it does not analyze the potential errors or scalability issues that may arise when integrating multiple units. As such, it is insufficient to convincingly support the practical feasibility and scalability of a full GMC array. Further experimental results and in-depth analysis are necessary to substantiate this claim.

Response 2-2: We appreciate the reviewer's insightful comment regarding the practical feasibility and scalability of the multiple GMC integrations. We acknowledge that our experiment mainly demonstrates functional feasibility, while a deeper analysis of the potential errors or scalability issues arising when integrating a larger number of units has not yet been fully addressed. Considering that array-level performance ultimately determines whether such devices can support large-scale CIM architectures in real-world applications, a quantitative evaluation of integration scalability is indeed essential.

As shown in Fig. R2a, given the intrinsic self-rectifying characteristics of the GMC devices, we employ the read margin as the key metric to characterize the scalability of the GMC array²³. The read margin reflects the ability of a memory array to accurately retrieve stored information under realistic conditions, including noise and sneak path currents. Therefore, it is a widely accepted and fundamental indicator for evaluating the scalability of self-rectifying memristive crossbar arrays.²³ Additionally the read voltage scheme during array operation has a significant impact on the sneak path current. The $V_{\text{read}}/2$ bias scheme is employed for current readout in this analysis ($V_{\text{read}} = 1.7 \text{ V}$).

Under this scheme, only the fully selected GMCs (with a read voltage of V_{read} , as indicated by pink dots in Fig. R2a) will have current, while the partially selected (with a read voltage of $V_{\text{read}}/2$, as indicated by green dots in Fig. R2a) and unselected GMCs (with a read voltage of 0, as indicated by blue dots in Fig. R2a) will have no current, effectively suppressing sneak-path currents.

Notably, the read margin typically decreases with integrated array size, and a margin below 10% is generally considered the practical scalability limit.²⁴ Experimentally, we evaluate the read margin using a single bit-line pull-up strategy, which is defined by the following relationships:²⁵

$$\text{Read Margin} = \frac{\Delta V}{V_{\text{pu}}} = R_{\text{pu}} \times \left(\frac{1}{R_1 + R_{\text{pu}}} - \frac{1}{R_2 + R_{\text{pu}}} \right) \quad (\text{R1})$$

$$\begin{cases} R_{\text{sneak}} = \frac{2R_{\text{pos}}}{n-1} + \frac{R_{\text{neg}}}{(n-1)^2} \\ R_1 = R_{\text{LRS}} \times \left(\frac{R_{\text{sneak}}}{R_{\text{LRS}} + R_{\text{sneak}}} \right) \\ R_2 = R_{\text{HRS}} \times \left(\frac{R_{\text{sneak}}}{R_{\text{HRS}} + R_{\text{sneak}}} \right) \end{cases} \quad (\text{R2})$$

Herein, ΔV refers to the difference between the input voltages of the selected cell at low-resistance state (LRS) and high-resistance state (HRS). V_{pu} denotes the pull-up voltage applied to the selected BL, which is represented by V_{read} in this experiment. R_{pu} is the pull-up resistance, and R_{sneak} is the resistance along the sneak paths. In addition, R_{pos} and R_{neg} refer to the resistance of a cell that is half selected in this experiment as well as a cell that is not selected at all, respectively. n represents the number of SLs or BLs in a whole crossbar array. As illustrated in Fig. R2b, with read margin as the sole parameter for evaluating the scalability of the proposed GMC array,

the array in this work can be scaled up to approximately 59×59 (3.48 Kb) operated at a read voltage of 1.7 V. This indicates that the GMC units proposed in this work can reliably perform read operations, demonstrating feasibility at the array level. Furthermore, the scalability of the GMC array could be further improved by increasing the self-rectifying ratio, for example, by increasing the read voltage or replacing other self-rectifying memristors with higher rectification ratios.^{23,24} It should be noted that the main goal of this work is to propose a potential hardware implementation of KAN, and future efforts will focus on further advancing the practical realization of hardware KAN.

Fig. R2 | Analysis of the array scalability. **a** Self-rectifying I - V characteristics of GMCs. **b** Scalability analysis based on the proposed GMCs with over 3.48 kb scale.

We have included the response in the latest Supplementary Note 3 and referenced it in the ‘Circuit design for GMC-based implementation of KANs’ section of the ‘Results’ in the latest main text. Correspondingly, Fig. R2 is presented as Supplementary Fig. 8.

3. The response includes an error modeling of device variability and analyzes its impact on task performance. However, it remains unclear whether the modeling aligns with actual physical behavior. To justify the validity of this approach, experimental validation using measured device variability is necessary.

Response 2-3: We are grateful for the reviewer's comment on the consistency between the previously adopted error modeling and the actual physical behavior of the GMCs, which is crucial for ensuring the reliability of the proposed physical G-KAN. The primary sources of errors in GMC units arise from device-to-device (D2D) variations (Fig. R3a) and cycle-to-cycle (C2C) variations within the same device (Fig. R3b). D2D variations are mainly caused by fluctuations in material defect density and channel area due to fabrication limitations^{26,27}, whereas C2C variations result from external random perturbations. Both types of errors manifest as non-idealities and inconsistencies in the quasi-Gaussian output of the GMC units.

To investigate the causes and nature of the random perturbations in the Gaussian-like transfer characteristic curve, a statistical analysis is required. Given that the Gaussian-like function is described by characteristic parameters such as the peak value, σ_+ and σ_- , error analysis can be conducted on the perturbations introduced in these parameters. Mathematically, these parameters can be defined as a class of variables that satisfy the following equation:

$$P^* = P(1 + KZ) \quad (\text{R3})$$

where the P^* denotes the experimentally measured value, Z is a random variable following a standard normal distribution (i.e., $Z \sim N(0,1)$), and K is defined as the

characteristic perturbation constant.²⁸ As an illustration, Fig. R3c shows the effect when $K = 0.1$.

Based on the above premise, to validate that the utilization of the coefficient of variation (CV) to quantify the parameter perturbation behavior is reasonable, it is sufficient to demonstrate that the CV is a variable explicitly related to the coefficient K in Eq. (R3). Through rigorous mathematical derivation, we have proved that the previously adopted error modeling based on the coefficient of variation (CV) is in fact equivalent to the above perturbation model. The detailed proof process is presented as follows.

Let P be the intrinsic value of a certain characteristic parameter, and let P_i ($i=1,2,3,\dots$) denote the measured value (random variable) of P at the i -th measurement. According to the perturbation model:

$$P_i = P(1 + KZ), Z \sim N(0,1) \quad (\text{R4})$$

Denote by $E(P)$ and $\text{Var}(P)$ the exception and variance of P , respectively. By exploiting the properties of the normal distribution, the following derivation is obtained:

$$E(P_i) = E(P) + KE(Z) = P \quad (\text{R5})$$

$$\text{Var}(P_i) = P^2 \text{Var}(1 + KZ) = P^2 \left\{ E \left[(1 + KZ)^2 \right] - \left[E(1 + KZ) \right]^2 \right\} \quad (\text{R6})$$

Since

$$E \left[(1 + KZ)^2 \right] = E(1 + 2KZ + K^2 Z^2) = 1 + K^2 E(Z^2) = 1 + K^2 \left\{ \text{Var}(Z) + \left[E(Z) \right]^2 \right\} = 1 + K^2$$

, and $E(1 + KZ) = 1$. Equation (R6) follows that,

$$\text{Var}(P_i) = K^2 P^2 \quad (\text{R7})$$

For the GMCs prepared in this work, $P > 0$. Therefore,

$$\frac{\sqrt{\text{Var}(P_i)}}{E(P_i)} = K, (K \geq 0) \quad (\text{R8})$$

According to the definition of the CV—the ratio of the standard deviation to the exception—the constant K is equivalent to CV. In summary, the previous error modeling based on CV is equivalent to the characterization perturbation model of GMCs. In combination with the laws of large numbers²⁹, the intrinsic value $P = E(P_i) \approx \frac{1}{n} \sum_{i=1}^n P_i$. Hence, this equivalence establishes a direct theoretical link between the error modeling framework and the statistical description of experimentally observed device variability. It is worth noting that, in the previously revised manuscript, to ensure the rationality of the perturbation simulation, each device model in a G-KAN was independently assigned its own characterization perturbation random variables, and the outcome was obtained through forward propagation at the network level, thereby reflecting the cumulative effect of the perturbations.

Fig. R3 | Device variability. **a** The transfer characteristic curves with 8 different GMCs. **b** Test of the transfer characteristic curve for 1000 cycles of the same GMC unit, among which the wine red color represents the output of the 500th cycle. **c** Random perturbations considered on the modeled GMC curve to emulate the impact of device

variability.

We have included the data in the 'Circuit Design for GMC-based Implementation of KANs' section of the 'Results' in the latest main text. Correspondingly, Fig. R3 is presented as Supplementary Fig. 5 and Supplementary Fig. 6.

4. The revised manuscript states that the RMSE of G-KAN remains below 0.4 when the factor q reaches 50. However, it is not evident what level of physical error this corresponds to in practice. Is such a quantization level sufficient in a real-world hardware scenario? A more detailed explanation is needed to substantiate the robustness of G-KAN under practical conditions.

Response 2-4: We sincerely thank the reviewer for raising this important comment regarding the physical significance of the factor q and correspondence to the error statistics in practical hardware scenarios.

In the previous version of the revised manuscript, q was introduced as a hypothetical scaling factor to magnify the intrinsic variability of the device by q -fold. This setting was applied to emulate the situation in which the intrinsic device variability is further amplified when the array operates under the influence of external noise sources. Specifically, the expression ‘ qC ’ refers to multiplying the CV of each characterization parameter. To avoid ambiguity, in the latest revised manuscript, we have clarified this description as “multiplying each intrinsic perturbation constant by q ”, which more clearly conveys the experimental details.

Regarding whether a q -fold perturbation level is sufficient for real-world hardware, we refer to the study by Li et al³⁰. In practice, in addition to the intrinsic device variability, array-level operation also suffers from perturbations introduced by peripheral circuits, such as sensing drift in the transimpedance amplifier (TIA).³⁰ The standard deviation of such drift has been reported to be approximately 0.3, which is still far below the case of 50-fold intrinsic variability (corresponding to

$P^* \sim N(P, 2500(KP)^2)$). Notably, when $q = 50$, the RMSE approximately reaches as high as 0.4. Although this represents a considerable error level for a 1D function regression task, it should be emphasized that such performance arises under extremely amplified perturbations—50 times the intrinsic device variation. The motivation for presenting this stress-test scenario was to benchmark against a M-MLP model with 15,250 weights (i.e., $152.5\times$ larger than G-KAN with $n_c = 100$). However, this M-MLP exhibited an RMSE as high as 1.311 even under ideal conditions. Hence, such extreme test was previously adopted to demonstrate that G-KAN could achieve much lower error than M-MLP under ideal conditions, even when subjected to such an extremely adverse testing premise. In realistic settings, GMC arrays operate under normal device-to-device interactions, where the perturbation factor q is expected to be close to 1. Even when $q = 2$ —already far beyond the typical operational conditions—the RMSE increases only by 0.0168, demonstrating that the system maintains strong robustness under practical circumstances.

Following the reasoning of Li et al.³⁰, if the network maintains robust performance (measured by AI benchmark metrics) even under artificially magnified error conditions (e.g., $q = 2$, which significantly exceeds the expected real-world variability), this strongly substantiates the reliability of the G-KAN in practical operation. Moreover, the underlying mechanism of neural network for classification is probabilistic decision-making, where the label with the highest probability is selected as the output.³¹ Under this principle, even if the predicted probabilities are affected by internal errors, the final classification outcome does not vary continuously as RMSE does in regression tasks,

but instead exhibits a certain degree of tolerance. Consequently, classification accuracy is less sensitive to perturbations. As shown in Fig. R4, even when $q = 5$, the accuracy decreases by only 0.453%. Therefore, G-KAN demonstrates strong robustness in both regression and classification tasks.

Fig. R4 | Robustness evaluation of G-KAN. The left panel presents the evaluation on the function regression task, while the right panel shows the evaluation on the image classification task.

We have included the response in latest Supplementary Note 7 and 9 and referenced it in the 'Circuit Design for GMC-based Implementation of KANs' section of the 'Results' in the latest main text. Correspondingly, Fig. R4 is presented as Fig. 5g in the latest main text.

5. The response mentions methods that allow the weighted coefficients of cells to be dynamically adjusted during training to compensate for effects such as current drift caused by device variations. However, these methods are designed primarily for conventional CNN or MLP architectures, which are fundamentally different from the proposed G-KAN. Whether such techniques are applicable to G-KAN requires further clarification.

Response 2-5: We sincerely appreciate the reviewer's insightful comments and acknowledge that a more in-depth discussion of this concern will help confirm the feasibility of the method within the architecture proposed in this work. Specifically, the method refers to in-situ training in neuromorphic computing architectures.^{30,32} Therefore, we first clarify the working mechanism of in-situ training in well-established systems such as MLPs and CNNs, as well as the details of how this method overcome device non-idealities. On this basis, we then analyze whether in-situ training remains feasible for G-KAN by considering the electrical write/erase mechanism of GMCs.

In-situ training is widely applied in various neuromorphic architectures that utilize memristors to store trainable parameters and operate on crossbar arrays. Examples include spiking neural networks (SNNs)³³, reservoir computing³⁴⁻³⁶, long short-term memory (LSTM) networks^{37,38}, and even hardware implementations of fast Fourier transforms (FFT)³⁹ and discrete Fourier transforms (DFT)⁴⁰. In these architectures, the trainable parameters are fundamentally stored in the conductance states of memristors, and the training process is accomplished by modulating these conductance values.⁴¹ Upon analyzing the GMC structure, which is composed of an anti-ambipolar transistor (AAT)⁴² in series with a memristor, the AAT solely provides the Gaussian-like transfer characteristics, while the memristor stores and modulate the trainable parameter of the Gaussian kernel. Similar to purely memristor-based networks, the in-situ training of G-KAN is also achieved by applying programming pulse sequences to adjust the memristor conductance. Moreover, the GMC is physically compatible with the crossbar array. Therefore, as a hardware-level optimization mechanism, in-situ training does not

depend on specific network architectures, and the KAN-type networks constructed with GMCs also inherently inherit the capability of in-situ training to overcome hardware non-idealities.

Secondly, regarding the role of in-situ training in overcoming current drift, further clarification is necessary. In fact, in-situ training not only effectively addresses current drift but also adaptively compensates for other various hardware non-idealities. Non-ideal factors such as current drift, write nonlinearity, device-to-device variation, wire resistance, and asymmetries in peripheral circuits typically affect the weight programming process.³² Moreover, in-situ training directly performs computation at the location where the neural network parameters are stored. This allows in-situ training to capture and compensate for these unavoidable device defects in real-time.³⁰

Specifically, in-situ training with backpropagation can adaptively adjust network parameters to minimize the impact of unavoidable hardware non-idealities without the need for any hardware prior knowledge. Therefore, in-situ training has the capability to automatically adjust weights and compensate for hardware defects.³⁰ In deeper networks, the adaptive ability of in-situ training is particularly pronounced because the hidden neurons can minimize the impact of hardware defects on network performance.³⁰ Experimental results from Li et al. have shown that in-situ training can effectively tolerate defects in other hardware components (e.g., transistors, peripheral circuits, and current sensing drift), further validating its robustness.³⁰ Especially in large-scale arrays, where some devices may fail to respond due to manufacturing or usage defects, in-situ training can continue to update the effective units without

significantly affecting the overall performance of the network. Even with 11% of devices being stuck and unresponsive, in-situ training maintains high performance, demonstrating its adaptive ability in the presence of hardware defects.³⁰ In summary, in-situ training is not merely a solution to current drift specific to MLPs and CNNs, but rather a device- and circuit-level strategy which is broadly feasible to the hardware compatible with crossbar array integration. This method enables neuromorphic circuits to effectively overcome various hardware non-idealities, thereby enhancing the performance and robustness of neural networks in actual hardware environments.

We have already included the explanations regarding the error analysis in Supplementary Note 9 and referenced this content in the 'Circuit Design for GMC-based Implementation of KANs' section of the 'Results' in the latest main text.

Reviewer #3 (Remarks to the Author):

The authors have conducted additional experiments and simulations that partially address my initial concerns regarding device measurements. However, the circuit designs presented in this manuscript remain underdeveloped and are insufficient to support even a design-level demonstration of computing-in-memory (CIM) for Kolmogorov - Arnold Networks (KAN), let alone an implementation of CIM-based KAN. Several significant concerns remain, even after the revisions, and the manuscript does not meet the standards required for publication in Nature Communications.

Response: We sincerely appreciate the reviewer for providing valuable feedback on the shortcomings in the circuit design and result presentation in the previous manuscript. The original manuscript indeed oversimplified some of the schematic details, which compromised the technical rigor. In response, we have invited a professor with extensive expertise in memristor chip design and fabrication to join the discussion. Through in-depth collaboration and multiple rounds of revisions, we have comprehensively and carefully improved the manuscript to effectively address the issues raised in the reviewer's comments. In detail, (1) we recognized that the initial circuit illustrations were adopted from overly simplified representations in several prior works^{10,43-45} and therefore lacked sufficient components; (2) we have now reconstructed the circuit schematics by referring to detailed circuit presentations in established neuromorphic computing literatures,^{10,30,46,47} and re-implemented the presentations on a professional electronic simulation platform to ensure accuracy and credibility; and (3)

we decomposed the previously identified design limitations into distinct sub-problems and proposed targeted solutions for each. Collectively, these efforts elevate professional circuit design which is adequate to support a computing-in-memory (CIM) demonstration for KANs.

1. The circuit designs presented in this manuscript are difficult to understand and appear unrealistic, which significantly undermines the authors' claim of implementing KAN using a CIM approach. The poor quality of the circuit design raises serious concerns about the credibility of the simulation results. Below are several fundamental questions that support this assessment:

Response 3-1:

1) What do the triangle symbols represent in inputs and outputs of Fig. 3a? What are their functions?

We appreciate the reviewer's attention to Fig. 3a. The triangle symbols were originally used as a simplified representation of differential operational amplifier stages, omitting the power supply pins and feedback paths. This simplification, which follows the convention adopted in some reports about memristive neural network, was intended to maintain clarity within the limited schematic space.^{10,45,48}

However, we acknowledge that such a representation may be easily confused with the comparator symbol commonly used in analog electronics. In response to the reviewer's comment, we have revised the figure to align with the standard depiction of output stages in memristive crossbar arrays. Specifically, the column currents from the

array are first converted into voltage signals via a transimpedance amplifier (TIA), which enables subsequent operations (Fig. R5). Since the non-inverting terminal is grounded, the virtual short property of the negative-feedback operational amplifier ensures that the potential of the entire SL column is maintained at zero.¹⁰ Then, the amplified differential signals can then be fed into a dedicated differential amplifier stage or a specialized processor for further processing. The specifics of these peripheral circuits are provided in the figure captions for clarity, as the ensuing discussion will focus on the core GMC array architecture.

Fig. R5 | Transimpedance amplifier (TIA).

2) What are the triangle symbols in Fig. 3b? Are they the same components as those in Fig. 3a?

In the Fig. 3b of original manuscript, the triangular symbols serve the same purpose as in the reviewer’s Question (1) – to simplify the representation in the block diagram. These symbols indicate differential operation, where “+” and “-” denote the non-inverting and inverting input terminals, respectively. To address the previous non-standard depiction, we have revised the figure and adopted the widely utilized TIA as the essential element at the MAC current output stage.

We sincerely apologize for the insufficient explanations and non-standard notations in our original submission, which may have caused confusion. Thank you again for pointing out these issues, which have helped us significantly improve the rigor and readability of the manuscript.

3) Fig. 1c shows that a GMC consists of a Gaussian transistor and a memristor. However, in Fig. 3b, memristors appear as independent devices when forming differential pairs. How is this structural change justified? What are the characteristics of these independent memristor differential pairs?

In the Fig. 3b of original manuscript, the standalone memristors (highlighted in gray) correspond to the synaptic weights of the residual connections in the KAN. These connections serve as additional pathways in the network structure, enabling the direct transmission of input information (Fig. R6).⁴⁹ In hardware implementation, these connections adopt exactly the same differential-pair encoding format for weights as conventional memristors. Through this design, the network can achieve more stable training and enhanced representational capability.⁴⁹

Fig. R6 | Information transfer between nodes—residual connection ($b(x)$) can be added on the basis function combination ($\sum_i c_i f_i(x)$).

This response is presented in Supplementary Note 4 and referenced in the 'Circuit Design for GMC-based Implementation of KANs' section of the 'Results' in the main

text. Correspondingly, Fig. R6 is combined and presented as Supplementary Fig. 12.

4) In Fig. 3c, what are the triangles with one terminal connected to x ? Where are the other terminals connected?

The triangle symbols used here represent a simplified version of the differential circuit as well, which was originally intended to achieve a horizontal translation of the function. In the previously revised manuscript, the other terminals were connected to external bias sources used to implement function translation, as illustrated in Fig. R7. Since a horizontal translation of the function is equivalent to a biasing of the independent variable (x), we utilized this translation property in differential form to achieve the effect shown in Fig. R8. Specifically, the transfer characteristics of the GMC are mapped to a laterally translated Gaussian-like function (Fig. R8a), which requires establishing a bias relationship between the input (x) and the gate voltage (v_g). In scenarios requiring higher fitting precision, such as the approximation of complex one-dimensional continuous functions, it is further necessary to contract the basis functions to form a finer grid (Fig. R8b). However, function contraction can be achieved simply by introducing a gain factor of k to the independent variable. In general, all these operations rely on addition/subtraction and subsequent amplification of x , which can be processed via operational circuits. Therefore, in the previously presented triangular symbols, the port other than the one connected to x represents the bias term. The other input of the triangle symbol corresponds to the applied bias term.

In the original manuscript, the bias terms were not explicitly labeled in the block diagram to avoid overcrowding the figure. Following the reviewer's suggestion, we

recognize that the original design may be confusing and lacks hardware-friendliness. As a result, we have re-evaluated and modified the configuration of the input terminals. Specifically, as shown in Fig. R8c, the contraction and translation of the Gaussian-like function essentially result from a fixed computational mode of linear encoding applied to the input variable. This operation can be realized simply by generating the gate voltages fed into the GMC array according to $v_g = k(x - a) + V_\mu$, which is inherently a peripheral process. Consequently, we eliminated the translation module in the original design, which unnecessarily increases hardware complexity, thereby optimizing the circuit design. The reconstructed figure can be found in Fig. R11.

Fig. R7 | Schematic illustration of the computational scheme for handling translation and scaling operations with external bias.

Fig. R8 | Function translation and device mapping. **a** Function translation induced by variable bias. **b** Function contraction induced by gain on the independent variable for finer-grid adaption. **c** The input signal (x) is embedded in the voltage applied to the gate electrode, and the mapped functional relationship is established through the output characteristics of the GMC: $f(x) = \exp\left[-\left(\frac{x-a}{\sigma}\right)^2\right]$.

This response is mentioned in the latest Supplementary Note 2 and referenced in the 'Circuit Design for GMC-based Implementation of KANs' section of the 'Results' in the latest main text. Correspondingly, Fig. R8 is presented as Supplementary Fig. 10 in the latest Supplementary Information.

5) In Supplementary Fig. 8, what is the 1T1R architecture, and is it the same as a GMC cell? The diagram includes numerous switches, what are their specific functions? Moreover, what is the difference between these switches and the MUXes, given that they are typically considered equivalent in circuit design?

In the Supplementary Fig. 8 of original manuscript, the specifically highlighted 1T1R structure represents the memristor differential pairs used for implementing the residual connection, distinguishing this design from the GMC configuration. In detail, in typical hardware implementations, a memristor-based crossbar array requires additional transistors to select specific rows or columns.¹⁰ This is because, in the absence of transistors, unintended operations may occur on unselected units within the two-dimensional arrays. Through the switching control of transistors, the 1T1R structure effectively prevents crosstalk between multiple memristor units, especially half-select interference. This is the main challenge in achieving high-precision tuning in passive integrated circuits.⁴⁵

In addition, the multiple switches shown in Supplementary Fig. 8 of the original manuscript function similarly to MUXes. This representation is referred to previously reported studies.⁵⁰ Specifically, these switches are utilized to select and connect different signal lines (VWL/VBL/VSL/GND) in various modes to alter the potential difference across the selected unit. For example, during feedforward progress, SLs are connected to GND, providing a zero-potential reference point. During backpropagation, if the selected unit requires a negative voltage, thence BLs are connected to GND while SLs are connected to VSL. The switching of WLs between VWL and GND are used to

select the row and column of the target unit. Additionally, the MUXes shown in original Supplementary Fig. 8 are used to implement the ReLU activation function. In the original Simulink module, these MUXes were implemented using the ‘Switch’ blocks (Fig. R9).

We sincerely apologize that, in the original design, directly paralleling the memristor component connected through the operational amplifier with other GMCs is inappropriate. This is because that, the operational amplifier forces the output potential V_{out} to $-R_F \cdot i$ (where R_F is the feedback resistor and i is the current flowing through the memristor), which contradicts the condition that the operational amplifier at the array’s total output enforces the SL column potential to be zero through the virtual short principle. Therefore, we correct the schematic, and the updated version is shown in Fig. R10. Specifically, since the memristor array and the GMC array adopt different electrical encoding schemes, they need to be allocated into two independent arrays. The corresponding output vectors from each array can then be summed to compute the residual-connected form of G-KAN, as illustrated in Fig. R10 and Fig. R11.

Fig. R9 | The ‘Switch’ block utilized in Simulink module to implement the ReLU function. When the ‘Input Signal’ is greater than zero, the block directly outputs the input; otherwise, this block outputs a ground signal (zero potential).

Fig. R10 | Hardware implementation of the G-KAN. **a, d** Topology of KAN. Unlike MLPs, each edge is not merely a simple synapse but can be expanded into an activation function represented by a combination of Gaussian-like basis functions (Gaussian-like kernels), as illustrated in **d**. **b** GMCs serve as the fundamental electrical units for implementing tunable Gaussian-like basis functions, and are compatible with crossbar array architectures to realize analog in-memory VMM and MAC operations. **c** Schematic of G-KAN. **e** Demonstration of function-level parallelism in a G-KAN during inference operations (m input nodes, p output nodes, with n basis functions between input-output node pair). **f** Basic performance evaluation of the G-KAN (with 100 basis functions), using a 1D function regression task as a representative example.

Fig. R11 | Schematic diagram of the extended G-KAN architecture with residual connections. **a** GMC array part. Each green block represents the combination of n_c Gaussian-like basis functions expanded from an input variable (x). **b** A 1T1R-standard integrated memristor array is employed to process the additional operations of the residual-connection part. **c** The GMC array and the $b(x)$ array are directly superimposed element-wise along the output vector to generate the final output.

In the schematics above, the ‘Subtractor Configuration’ contains the peripheral circuits utilized to handle the differential operation of the output voltages, which can be implemented using an operational amplifier (op-amp) (Fig. R12). Specifically, two TIAs as a differential pair convert the current representing the MAC operation result on the SL column into voltage, and then perform the differential calculation through an op-amp. The electrical principles in Fig. R12 can be found in following description.

For the non-inverting terminal, according to the op-amp’s virtual open and KCL, the result follows that:

$$\frac{V_2 - V_+}{R_1} = \frac{V_+ - 0}{R_2} \Rightarrow V_+ = \frac{V_2}{1 + \frac{R_1}{R_2}} \quad (\text{R9})$$

Similarly, for the inverting terminal, the result can be derived that:

$$\frac{V_1 - V_-}{R_1} = \frac{V_- - V_o}{R_2} \Rightarrow V_- = \frac{V_1 + \frac{R_1}{R_2} V_o}{1 + \frac{R_1}{R_2}} \quad (\text{R10})$$

By also considering the virtual short characteristic ($V_+ = V_-$) of the op-amp, and combining Eq. R9 and R10, the final output (V_o) is exactly the result of the differential

operation: $V_o = \frac{R_2}{R_1}(V_2 - V_1)$.

Fig. R12 | An op-amp in the ‘Subtractor Configuration’, used to perform the differential operation of the voltages.

The response is presented in Supplementary Note 2 of the latest revised manuscript and is referenced in the 'Circuit Design for GMC-based Implementation of KANs' section of the 'Results' in the main text. Correspondingly, Fig. R10 is presented as Fig.

4 in the main text, Fig. R11 as Supplementary Fig. 9, and Fig. R12 as Supplementary Fig. 11.

6) In Supplementary Fig. 9, a ReLU function cannot be implemented with only a single analog switch. How is the comparison operation (i.e., determining whether $x < 0$ or $x > 0$) realized in the analog domain?

Thank you for your in-depth review and valuable comments on this detail. Indeed, it is not feasible to implement the rectification output of the ReLU function using only a single switch in the analog domain. The single switch symbol in the previous illustration was intended to facilitate the understanding of the basic function of a ReLU-like electronic switch. However, since this form lacks the ability to determine the polarity of the input voltage, a detailed demonstration of the ReLU activation function is provided here.

Firstly, to provide a more intuitive comparison of the hardware implementation complexity between the SiLU and ReLU nonlinear activation functions, we present the complete circuit schematics of both the SiLU⁵¹ and ReLU designs (Fig. R13). Specifically, the construction of the SiLU circuit⁵¹ is relatively challenging, as this circuit involves an exponential circuit (highlighted in the orange box), an external constant current source (1 A), two operational amplifiers, and a division circuit. In particular, the exponential circuit additionally requires a constant current source supplying the reference current and two operational amplifiers.⁵¹ Under these conditions, configuring such a complex circuit merely to generate the output of a single SiLU(x) function is highly impractical for large-scale network integration.

On the other hand, the ReLU function ($\text{ReLU}(V_i) = \max\{0, V_i\}$) can be implemented through a combination of the feedback amplifiers and diodes to achieve analog-level transformation (Fig. R14a). This circuit has been validated through Simscape electrical simulations, and the ReLU operation is well-supported by the simulation result (Fig. R14b and c). This approach is designed to transmit the voltage in the ReLU form and effectively determine the polarity of the input signals.

Fig. R13 | Hardware implementation of SiLU and ReLU functions. **a** Graph of SiLU(x) . **b** Graph of ReLU(x) . **c** Schematic of SiLU circuit. **d** Schematic of ReLU circuit.

Fig. R14 | Rectifying linear unit (ReLU) circuit. **a** Circuit schematic. V_i and V_o represent the input and output voltages of the ReLU function circuit module, respectively. **b** Circuit built within the Simscape electrical simulation platform. For the configuration, V_i is swept from -1 V to 1 V using a controlled voltage source, and the V_o is measured using a voltage sensor. **c** $V_o - V_i$ characteristic curve, which is consistent with the numerical simulation of ReLU.

Additionally, considering that the ReLU function at the input side processes the

given analog signals rather than performing computations within the network, a simpler approach is to directly utilize a signal function generator (included in the input buffer⁵⁰) to generate the required signals. However, this part pertains to the peripheral configuration for generating input signals, rather than the processing element (PE) chip⁵² responsible for processing within the neural network.⁵³ Therefore, such peripheral configuration falls outside the scope of this study. The specific simplified approach to be used can be chosen based on the actual PE board design.

The response is mentioned in Supplementary Note 10 of the latest revised manuscript and is referenced in the 'Principle of MAC Operations in G-KANs' section of the 'Methods' in the main text. Correspondingly, Fig. R13 is presented as Supplementary Fig. 20, and Fig. R14 as Supplementary Fig. 21.

7) Simulink is generally used for system-level simulations, and VHDL is designed for modeling digital circuits at the behavioral level. Since the proposed CIM framework targets fully analog circuits, the simulation results shown in Supplementary Fig. 10 are not representative of analog behavior and are therefore unconvincing.

We sincerely appreciate the reviewer's critical insight regarding the appropriateness of simulation tools used in Supplementary Fig. 10 of the original manuscript. We acknowledge that the earlier description may have caused misunderstanding, and we regret any confusion this may have caused.

To clarify, the previously mentioned reference to 'VHDL' stemmed from the optional settings under 'HDL Code Generation > Set Basic Options' within Simulink. It was not our intention to suggest that VHDL was directly involved in the simulation

or modeling process. The actual model presented in the previous Supplementary Fig. 10 was constructed entirely using the ‘Commonly Used Blocks’ library in Simulink. Notably, each GMC was encapsulated as a self-contained functional module, with input-output behavior designed to reflect experimentally measured $I_d - V_g$ characteristics of the fabricated devices in this work.

To address the reviewer’s concern about analog-level validation, we have revised the simulation methodology in the current version. Specifically, we reconstructed the full array of 200 GMCs in the Simscape Electrical platform (‘Foundation Library > Electrical’) to ensure more rigorous physical-level modeling. As illustrated in Fig. R15a, a single GMC unit—composed of an AAT and a memristor in series—cannot be instantiated directly from standard Simscape library components. However, based on GMC’s characteristic behavior, each GMC branch can be accurately modeled as a voltage-controlled current source (VCCS). Subsequently, following Kirchoff’s current law (KCL), the current output from a parallel array of GMCs (Fig. R15b) equals the algebraic sum of all individual branch currents. This cumulative current inherently performs the MAC operation across all basis functions in the analog domain, fully capturing the kernel computation of the KAN architecture. The resulting current is then converted to an analog voltage via a transimpedance amplifier (TIA). Such a configuration utilizes the negative feedback of the operational amplifier to form a virtual ground, thereby clamping the source line voltage to a stable ground level and minimizing readout errors.¹⁰ Finally, the differential operation is performed by an op-amp (Fig. R15c). Notably, the entire forward computation process—covering input

application, weighted current generation, voltage readout, and differential-pair subtraction—is completed entirely in the analog domain within the full array (Fig. R15d). In addition, Fig. R16 displays the oscilloscope waveform captured during the operation of this circuit.

Fig. R15 | GMC array constructed in the Simscape Electrical platform. a VCCS equivalent of the GMC branch and the parallel equivalent circuit of the GMC array. **b** Partial view of the crossbar circuit of GMCs. **c** TIAs used for voltage-mode output ($R=1\Omega$), while the op-amp for differential operation with $R_1=(I_M - I_m)\Omega$ and $R_2=1\Omega$. Here, I_M and I_m represent the values of the GMC current peak,

respectively. **d** Demonstration of the GMC array with 100 basis functions (200 GMCs).

Furthermore, the output voltages of a pair of TIAs are processed through an op-amp to

perform an amplified differential operation to produce the final output ($V_o = \frac{I^+ - I^-}{I_M - I_m}$)

of the G-KAN's network computation. The output waveform is then observed using a

'Voltage Sensor' and a 'Scope' block.

Fig. R16 | The oscilloscope waveform of the output voltage sweep for the circuit corresponding to Fig. R15. This measurement used a sampling period of 0.02 seconds, collecting a total of 5000 output signals via a voltage sensor. The horizontal axis represents time (in seconds), and the vertical axis represents voltage (in volts).

In summary, these enhancements not only strengthen the physical fidelity of the simulation but also more accurately reflect the analog behavior of the proposed CIM framework. We thank the reviewer again for this valuable suggestion, which has led to a significant improvement in the rigor and clarify of our work.

The response is mentioned in Supplementary Note 2 of the latest revised manuscript and is referenced in the 'Circuit Design for GMC-based Implementation of

KANs' section of the 'Results' in the main text. Correspondingly, Fig. R15 is presented as Supplementary Fig. 13.

Response 3-2:

(2.1) The claimed advantages of implementing KANs using GMC-based CIM are not supported by solid evidence. The algorithm-level benefits highlighted by the authors are inherent to the KAN architecture itself and are not specific to the proposed hardware implementation.

We sincerely thank the reviewer for the valuable insights into the integration advantages of KAN and GMCs. This issue is fundamental to the core intent to this research. We note that Reviewer 2 has independently addressed this central point, which further convinces us that providing a detailed explanation of this aspect is crucial to highlighting the value of our work. Therefore, we offer this comprehensive response to substantiate the key contributions of this research.

As Gaussian kernels (i.e., the simplified description of 'Gaussian-like basis functions') are the fundamental computational elements in KANs, the effectiveness of KAN's hardware implementation largely depends on how efficiently the computation and training of these kernels can be realized through devices or circuits. Existing hardware architectures, including von Neumann and near-memory systems, encounter significant limitations both in executing Gaussian kernels and in performing training, which restricts the full potential of KANs. The proposed GMCs are introduced specifically to address the shortcomings that current devices and architectures cannot overcome (Fig. R1). To emphasize the transformative impact of GMC-based

implementation on hardware KANs, the following content will provide real-world examples that illustrate why traditional hardware architectures are inadequate for KAN's implementation, thereby highlighting the advantages of G-KANs.

(1) Dilemmas of von Neumann architecture (corresponding to Dilemma 1-3 in Fig. R1b)

In von Neumann architectures, the physical separation between the processor and memory leads to the continuous transfer of data and instructions between them during computation due to the disparity between memory bandwidth and awaiting data.¹ Moreover, memory access introduces significant time and energy overhead, with much of the energy being consumed during data transfer rather than computation (**Dilemma 1 and 2**).^{1,2} The graphics processing unit (GPU), a prime example of optimized computing design, is widely used due to its exceptional performance in general matrix multiplication (GEMM)³. GPUs excel in neural network computations, efficiently performing linear transformations with high computational density by utilizing regular operations and contiguous memory access⁴, while the central processing unit (CPU) merely initiates scheduling, minimizing offloading overhead.⁵ However, these capabilities are not fully realized in KAN's computations. Using Gaussian kernels as basis functions, the inference of a KAN requires sequential operations such as squaring ($(x-a)^2$), scaling ($-\frac{1}{\sigma} \cdot (x-a)^2$), exponential transformation ($\exp(\cdot)$), weighted multiplication ($c_i \cdot \exp(\cdot)$), and accumulation (\sum). These operations cannot be directly handled by GEMM, necessitating frequent kernel launches during each feedforward pass, with each launch scheduled by the CPU, thus significantly increasing

the CPU-GPU offloading overhead (**Dilemma 3**). Even more concerning, KAN's more complex computational elements require more hardware resources and power than MLPs, even when the number of parameters is the same.^{2,6}

(2) Dilemmas of near-memory implementations (corresponding to Dilemma 4-6 in Fig. R1b)

The previous evidence highlights the incompatibility of KANs with the von Neumann architecture. As a result, it is essential to implement KANs on hardware platforms with greater computational and memory density. In this context, researchers have sought to overcome the limitations of the von Neumann architecture through near-memory computing.⁷ For KAN-related applications, typical approaches include lookup table (LUT)⁸ and piecewise linear (PWL)⁹ circuit designs. Although LUT-based circuits can output Gaussian kernels directly at the hardware level, this approach introduces considerable computational complexity and storage overhead. For instance, implementing a single Gaussian kernel requires 452 LUTs.⁸ When extended to network-level implementations, particularly in densely integrated architectures like KANs, the resulting operational costs become unsustainable (**Dilemma 4**).

Furthermore, recent research has applied CMOS integration technology based on PWL approximation to model activation functions in KANs.⁹ However, this approach presents several fundamental limitations: First, the PWL function requires predefined breakpoints (X_n), slopes (M_n), and intercepts (Y_n) for each segment ($X_n \leq X < X_{n+1}$), which must be stored in external memory. Second, the hardware architecture must incorporate numerous comparators (COMPs) and sensing circuits to determine the

segment corresponding to the input signal, retrieve the associated parameters (M_n and Y_n) from memory, and pass them to the MAC circuits to complete the linear computation. As such, even for simple operations, this method still incurs significant hardware resource overhead (**Dilemma 5**).

An overall assessment of both LUT-based and PWL-based strategies reveals that not only do they demand considerable hardware resources—such as multipliers (MPLs), adders (ADDs), and comparators (COMPs)—for executing Gaussian kernel computations, but also, due to their hardwired architecture, they are incapable of performing parameter updates (**Dilemma 6**). Although both strategies attempt to mitigate the von Neumann bottleneck, they ultimately fall short due to their inability to incorporate trainable characteristics into the neural networks.

Given these limitations in existing CMOS-based implementations, there has been a shift towards the exploration of dedicated KAN-type CIM architectures. Clearly, memristors alone are insufficient to perform in-memory computation for nonlinear functions like Gaussian kernels. The following section will specifically discuss how the GMCs and array design introduced in this work effectively address these challenges.

(3) Solutions leveraging GMC-based implementation (corresponding to Solution 1-5 in Fig. R1b)

The transfer characteristics of GMCs enable the direct generation of stable, parameterized Gaussian kernels, effectively capturing the mathematical behavior of Gaussian-like functions within the analog domain. Additionally, by adjusting the memristor conductance via write voltages, the peak of the Gaussian kernel can be

flexibly controlled, allowing electrical programming of the learnable coefficients. On the architectural side, the one-transistor-one-memristor structure of GMCs facilitates seamless integration into crossbar arrays.¹⁰ According to KCL, the output currents from all GMCs in the same column can be physically summed on the source line, directly performing the linear combination of Gaussian kernels in a KAN. Consequently, the GMC-based implementation enables the full computation from input to output to be completed within the analog domain (**Solution 1**).

Moreover, the traditional dilemma in CMOS-based implementations (**Dilemma 2**) arises from the physical separation of processor and memory.¹ The GMC-based solution fundamentally alters this model by adopting the CIM paradigm. The core advantage of this approach is that computation and data storage are synchronized at the same physical location, eliminating the need for frequent data transfers between memory and processing units. This removes the conflict between memory bandwidth and processing throughput, and concentrates energy consumption on computation (**Solution 2**). At the same time, the in-situ CIM operation mode completely avoids the offloading overhead intrinsic to the von Neumann architecture (**Solution 3**).

To address the hardware scaling challenge (**Dilemma 4**) faced by CMOS-based systems—where increasing the number of basis functions leads to a nonlinear explosion in hardware resource consumption—the GMCs introduced in this study provide a straightforward solution through linear scaling approach. Specifically, under the differential-pair encoding scheme, each Gaussian kernel requires only a fixed pair of GMCs. This ensures that hardware scaling grows linearly with the effectively prevents

unmanageable resource consumption, keeping the hardware scale of a G-KAN predictable and scalable (**Solution 4**).

Equally important, the GMC-based approach avoids the complex and segmented operations typical of near-memory computing solutions (Dilemma 5 and 6). Within GMC arrays, an input vector is loaded in parallel across all rows, enabling each cell to simultaneously perform in-situ Gaussian-like mapping and MAC operations. By utilizing the native physics of the crossbar array and KCL, the entire matrix transformation is executed in one step, eliminating the need for piecewise function evaluation, memory-based parameter retrieval, and data transfer. This one-step operation completely removes the timing and control overhead associated with multi-step processes in CMOS-based implementations, significantly reducing computational complexity (**Solution 5**).

The main dilemmas faced by the CMOS-based implementations, along with their corresponding solutions provided by the GMC-based approach, are visually summarized in **Fig. R1**. In addition, a comprehensive comparison of the key features of the reference objects is presented in **Table R1**.

Fig. R1 | Overview of KAN's hardware implementation strategies. a Reference object and benchmark metrics for the hardware implementation of KAN. **b** The dilemmas encountered by the von Neumann and near-memory architectures in processing the computations of KAN, while the GMC-based implementation provides a reliable solution at the in-memory computing level. **c** The GMC ingeniously combines the Gaussian kernel provided by AAT with the parameterization capability of memristors, integrating the necessary physical foundation for the in-memory computing required by KAN.

Furthermore, to systematically underscore the advantages of GMCs in implementing KAN and in representing the basis functions (i.e., Gaussian kernels)

associated with each activation link, we provide a comparative analysis against alternative implementation routes along with their corresponding reference objects and benchmark metrics. **Table R1** offers a comprehensive summary across multiple dimensions of comparison, as outlined below.

Table R1 Comparison for implementing KAN across different benchmarks.

Reference objects	Von Neumann Architecture	Near-Memory Architecture	GMC-based Crossbar Array
Parameterized Gaussian kernel implementation	 1. hardware dependance: DSPs/GPUs/FFs/RAMs/LUTs^{6,11-13}; 2. Muti-step pointwise operators (squaring→multiplication→exp()→multiplication); 3. memory-bound,^{14,15} 4. frequent memory accessing¹⁶ 	 1. memory-bound; 2. Separation of parameterization and kernel computation; 3. Non-adjustable functions (hardwired); 4. Reliance on a large number of computational modules^{8,9} 	physically in-device generation
if one-shot?	No	No	Yes
Computing in memory	NOT	partial	complete
if one-shot?	No	No	Yes
Pipeline	 1. Data loading and scheduling initialization (CPU side); ↓ 2. Multi-step chain of pointwise operators; ↓ 	(LUT-based)⁸ ADC→multi-bit input (address)→multi-bit output→DAC; (PWL-based)⁹ binary encoding→breakpoint detection→accessing	analog MAC implemented on the crossbar array

	3. Weighted accumulation; ↓ 4. Memory copy ☹️	slope/intercept memory →parameter fetching→digital MAC→DAC ☹️	☺️
if one-shot?	No	No	Yes
Computational parallelism	(partially parallel on GPUs) 1. memory-bound;¹⁷ 2. high overhead of data movement;¹⁸ 3. significant CPU-GPU offloading latency and energy consumption;¹⁹ 4. high overhead of scheduling and synchronization^{20,21} ☹️	(partially parallel)⁹ 1. memory-bound; 2. massive computational modules; 3. intricate control flow (PWL-based scene); 4. redundancy in address (LUT-based scene) or parameter (PWL-based scene) access ☹️	(completely parallel) 1. access-free memory; 2. offloading-free; 3. one-shot parallel VMM and MAC process; 4. simple control mechanism (voltage-pulse direct drive) ☺️
Hardware overhead change	1. memory and computation demand grows superlinearly with the number of parameters; 2. exponentially increasing memory copy latency;[*] 3. memory bandwidth bottleneck amplifies with increasing computational scale²² ☹️	1. memory and computation demand grows superlinearly with the number of parameters ☹️	1. device count demand increases linearly with computational scale (one basis function is implemented by a pair of GMCs) ☺️
In-situ training	--	NO (hardwired)	YES

			--	--	---	---

* <https://developer.nvidia.com/gdrcopy>

We have included the response in Supplementary Note 1 of the latest revision of the manuscript. Additionally, Fig. R1 is presented as Fig. 2 in the latest main text, while Table R1 is shown as Supplementary Table 1 in the latest Supplementary Information.

(2.2) At the hardware level, the claim of reduced device count compared to MLPs is based on an unfair comparison: the authors only consider the number of memristor devices required for KANs versus MLPs, while overlooking the significantly higher circuit complexity associated with implementing KANs using GMCs. This undermines the validity of the claimed hardware efficiency.

Thanks for the reviewer's critical observation. We have discussed about the reviewer's comment regarding the comparison of device counts. Since a complete hardware system involves numerous components, and the peripheral design methodologies of MLPs and KANs differ in ways that make their device counts hard to quantify directly, a simple comparison based solely on memory devices would be unfair. We sincerely appreciate the reviewer for highlighting this critical point, and we have accordingly removed this experiment from the manuscript to ensure the overall rigor of this work.

Reference

- 1 Vijaykumar, N. *et al.* A framework for accelerating bottlenecks in GPU execution with assist warps. *arXiv preprint arXiv:1602.01348* (2016).
- 2 Maurya, A., Rafique, M. M., Cappello, F. & Nicolae, B. MLP-Offload: Multi-Level, Multi-Path Offloading for LLM Pre-training to Break the GPU Memory Wall. *arXiv preprint arXiv:2509.02480* (2025).
- 3 Ansari, M. Q. & Ansari, M. Q. Accelerating Matrix Multiplication: A Performance Comparison Between Multi-Core CPU and GPU. *arXiv preprint arXiv:2507.19723* (2025).
- 4 Ho, K., Zhao, H., Jog, A. & Mohanty, S. "Improving gpu throughput through parallel execution using tensor cores and cuda cores," in *2022 IEEE Computer Society Annual Symposium on VLSI (ISVLSI)*. 223-228 (IEEE).
- 5 Driss, S. B., Soua, M., Kachouri, R. & Akil, M. "A comparison study between MLP and convolutional neural network models for character recognition," in *Real-Time Image and Video Processing 2017*. 32-42 (SPIE).
- 6 Le, T. X. H. *et al.* "Exploring the limitations of kolmogorov-arnold networks in classification: Insights to software training and hardware implementation," in *2024 Twelfth International Symposium on Computing and Networking Workshops (CANDARW)*. 110-116 (IEEE).
- 7 Singh, G. *et al.* "A review of near-memory computing architectures: Opportunities and challenges," in *2018 21st Euromicro Conference on Digital System Design (DSD)*. 608-617 (IEEE).
- 8 Shymkovych, V., Telenyk, S. & Kravets, P. Hardware implementation of radial-basis neural networks with Gaussian activation functions on FPGA. *Neural Computing and Applications* **33**, 9467-9479 (2021).
- 9 Sudarshan, C., Manea, P. & Strachan, J. P. A Kolmogorov–Arnold Compute-in-Memory (KA-CIM) Hardware Accelerator with High Energy Efficiency and Flexibility. (2025).
- 10 Aguirre, F. *et al.* Hardware implementation of memristor-based artificial neural networks. *Nature communications* **15**, 1974 (2024).
- 11 Cintra, R. *et al.* Gaussian kernel approximations require only bit-shifts. *Information* **15**, 618 (2024).
- 12 Huang, W.-H. *et al.* "Hardware acceleration of kolmogorov-arnold network (kan) for lightweight edge inference," in *Proceedings of the 30th Asia and South Pacific Design Automation Conference*. 693-699.
- 13 Mammadzada, F. Design of a Kolmogorov-Arnold Network Hardware Accelerator. (2025).
- 14 Gungor, M. *Optimizing the use of different memory types on modern fpgas*, Northeastern University, (2024).
- 15 Alavani, G., Desai, J., Saha, S. & Sarkar, S. Program analysis and machine learning–based approach to predict power consumption of cuda kernel. *ACM Transactions on Modeling and Performance Evaluation of Computing Systems* **8**, 1-24 (2023).
- 16 Lam, D. & Wunsch, D. Unsupervised feature learning classification with radial

- basis function extreme learning machine using graphic processors. *IEEE transactions on cybernetics* **47**, 224-231 (2016).
- 17 Wang, Y. E., Wei, G.-Y. & Brooks, D. Benchmarking TPU, GPU, and CPU
platforms for deep learning. *arXiv preprint arXiv:1907.10701* (2019).
- 18 Xu, W. *et al.* "ScaleDNN: data movement aware DNN training on multi-GPU,"
in *2021 IEEE/ACM International Conference On Computer Aided Design
(ICCAD)*. 1-9 (IEEE).
- 19 Lustig, D. & Martonosi, M. "Reducing GPU offload latency via fine-grained
CPU-GPU synchronization," in *2013 IEEE 19th International Symposium on
High Performance Computer Architecture (HPCA)*. 354-365 (IEEE).
- 20 Dev, K. & Reda, S. "Scheduling challenges and opportunities in integrated cpu+
gpu processors," in *Proceedings of the 14th ACM/IEEE Symposium on
Embedded Systems for Real-Time Multimedia*. 78-83.
- 21 Feng, W.-c. & Xiao, S. "To GPU synchronize or not GPU synchronize?," in *2010
IEEE International Symposium on Circuits and Systems (ISCAS)*. 3801-3804
(IEEE).
- 22 Ojika, D. *et al.* Addressing the memory bottleneck in AI model training. *arXiv
preprint arXiv:2003.08732* (2020).
- 23 Zhang, G. *et al.* Self-rectifying memristors with high rectification ratio for
attack-resilient autonomous driving systems. *Nature Communications* **16**, 5759
(2025).
- 24 Jeon, K. *et al.* Purely self-rectifying memristor-based passive crossbar array for
artificial neural network accelerators. *Nature communications* **15**, 129 (2024).
- 25 Ren, S.-G. *et al.* Pt/Al₂O₃/TaO_x/Ta self-rectifying memristor with record-
low operation current (< 2 pA), low power (fJ), and high scalability. *IEEE
Transactions on Electron Devices* **69**, 838-842 (2021).
- 26 Li, S. *et al.* Wafer - scale 2D hafnium diselenide based memristor crossbar array
for energy - efficient neural network hardware. *Advanced Materials* **34**,
2103376 (2022).
- 27 Feng, X. *et al.* Self-selective multi-terminal memtransistor crossbar array for in-
memory computing. *ACS nano* **15**, 1764-1774 (2021).
- 28 Lee, C. *et al.* Highly parallel and ultra-low-power probabilistic reasoning with
programmable gaussian-like memory transistors. *Nature Communications* **15**,
2439 (2024).
- 29 Révész, P. *The laws of large numbers*. Vol. 4 (Academic Press, 2014).
- 30 Li, C. *et al.* Efficient and self-adaptive in-situ learning in multilayer memristor
neural networks. *Nature communications* **9**, 2385 (2018).
- 31 Bishop, C. M. & Nasrabadi, N. M. *Pattern recognition and machine learning*.
Vol. 4 (Springer, 2006).
- 32 Wang, Z. *et al.* In situ training of feed-forward and recurrent convolutional
memristor networks. *Nature Machine Intelligence* **1**, 434-442 (2019).
- 33 Zheng, N. & Mazumder, P. Learning in memristor crossbar-based spiking neural
networks through modulation of weight-dependent spike-timing-dependent
plasticity. *IEEE Transactions on Nanotechnology* **17**, 520-532 (2018).

- 34 Jang, Y. H. *et al.* Spatiotemporal data processing with memristor crossbar -
array - based graph reservoir. *Advanced Materials* **36**, 2309314 (2024).
- 35 Zhong, Y. *et al.* Dynamic memristor-based reservoir computing for high-
efficiency temporal signal processing. *Nature communications* **12**, 408 (2021).
- 36 Zhong, Y. *et al.* A memristor-based analogue reservoir computing system for
real-time and power-efficient signal processing. *Nature Electronics* **5**, 672-681
(2022).
- 37 Li, C. *et al.* Long short-term memory networks in memristor crossbar arrays.
Nature Machine Intelligence **1**, 49-57 (2019).
- 38 Dou, G., Zhao, K., Guo, M. & Mou, J. Memristor-based LSTM network for text
classification. *Fractals* **31**, 2340040 (2023).
- 39 Xing, Y., Deng, M., Li, Z., Wang, Y. & Li, Q. "A Memristor-Based Winograd
Fast Fourier Transform Computation Circuit," in *2024 4th International
Conference on Electronic Information Engineering and Computer (EIECT)*.
13-17 (IEEE).
- 40 Zhao, H. *et al.* Memristor-based signal processing for edge computing. *Tsinghua
Science and Technology* **27**, 455-471 (2021).
- 41 Alibart, F., Zamanidoost, E. & Strukov, D. B. Pattern classification by
memristive crossbar circuits using ex situ and in situ training. *Nature
communications* **4**, 2072 (2013).
- 42 Lv, Y. *et al.* Robust Anti-Ambipolar Behavior and Gate-Tunable Rectifying
Effect in van der Waals p-n Junctions. *ACS Applied Electronic Materials* **4**,
5487-5497 (2022).
- 43 Zhang, W. *et al.* Hardware - friendly stochastic and adaptive learning in
memristor convolutional neural networks. *Advanced Intelligent Systems* **3**,
2100041 (2021).
- 44 Kawahara, T. & Mizuno, H. *Green computing with emerging memory*.
(Springer, 2013).
- 45 Kim, H., Mahmoodi, M. R., Nili, H. & Strukov, D. B. 4K-memristor analog-
grade passive crossbar circuit. *Nature communications* **12**, 5198 (2021).
- 46 Bhat, N. & Goswami, S. Neuromorphic pathways for transforming AI hardware.
Nature Electronics, 1-5 (2025).
- 47 Lin, P. *et al.* Three-dimensional memristor circuits as complex neural networks.
Nature Electronics **3**, 225-232 (2020).
- 48 Yamaoka, M. in *Green Computing with Emerging Memory: Low-Power
Computation for Social Innovation* 59-85 (Springer, 2012).
- 49 Liu, Z. *et al.* Kan: Kolmogorov-arnold networks. *arXiv preprint
arXiv:2404.19756* (2024).
- 50 Zhang, W. *et al.* Edge learning using a fully integrated neuro-inspired memristor
chip. *Science* **381**, 1205-1211 (2023).
- 51 Yang, C., Wang, X. & Zeng, Z. Full-circuit implementation of transformer
network based on memristor. *IEEE Transactions on Circuits and Systems I:
Regular Papers* **69**, 1395-1407 (2022).
- 52 Yao, P. *et al.* Fully hardware-implemented memristor convolutional neural

- network. *Nature* **577**, 641-646 (2020).
- 53 Kiani, F., Yin, J., Wang, Z., Yang, J. J. & Xia, Q. A fully hardware-based memristive multilayer neural network. *Science advances* **7**, eabj4801 (2021).

REVIEWER COMMENTS

Reviewer #2 (Remarks to the Author):

I appreciate the authors' comprehensive responses, which have satisfactorily addressed all my previous concerns. Therefore, I recommend the manuscript for publication.

We sincerely appreciate your recognition of this research and the revisions. The critical comments you previously provided played a key role in refining the logic and enhancing the rigor of this paper. We are very pleased that the revisions have met your approval, and we deeply appreciate the time and effort you dedicated to the entire review process!

Reviewer #3 (Remarks to the Author):

The authors have thoroughly revised both the manuscript and the supplementary information, addressing most of my previous concerns regarding peripheral circuit designs. While the manuscript has improved in technical details, I still have concerns about the presented comparisons and several major claims. As I understand it, the primary contributions of this work are the fabrication and characterization of the GMC cell, along with circuit design and simulation demonstrating the feasibility of using the GMC cell for hardware implementation of KAN.

Response:

We appreciate your recognition of the revisions made to the proposed circuit design. Under your guidance, we have addressed key weakness in the previous version, thereby significantly enhancing the scientific rigor of this research. Meanwhile, we fully understand the reviewer's cautious stance on the current technical details and main conclusions. In accordance with your suggestions, we have further revised the manuscript and included analysis of relevant hardware benchmark metrics to more robustly support the conclusions of this study regarding hardware advantages.

1. The authors claim that the GMC-based KAN exhibits “strong learning capabilities” and “excellent continual learning capability.” These properties are inherent to the KAN architecture and would be expected regardless of whether the GMC cell is used. I therefore recommend that the authors clearly distinguish algorithmic capabilities from device-level contributions. In particular, the manuscript should emphasize the

advantages introduced by the GMC cell, such as improved energy efficiency and computational speed enabled by in-memory computing, rather than implying that these learning capabilities originate from the GMC cells.

Response to 3-1:

We appreciate your constructive comments regarding the appropriateness of the conclusion statements. Following your suggestions, we have revised the relevant sections in the main text, including the Abstract, Introduction, Results, Discussion, as well as related statements in Supplementary Information (marked in red). These revisions aim to clearly state that the simulation results of the architecture proposed in this work is intended to validate the inheritance to KAN's advantages (which is the prerequisite for the system's feasibility), rather than relying on GMCs themselves to achieve these algorithmic advantages. In addition, the revised phrasing aims to avoid any overclaimed expressions regarding the contributions of this study. Furthermore, regarding the parallel computing advantages and energy efficiency improvements brought by the computing-in-memory architecture, we have provided a detailed explanation and quantitative study in the response to next question.

2. The performance comparison between G-KAN and M-MLP in Fig. 5 is not directly relevant to the main contributions of the work. As I understand it, a comparison between KAN and MLP would yield similar trends, since the observed performance differences are attributable to the underlying network architectures rather than the use of the GMC cell. Therefore, the advantages shown in Fig. 5 do not arise from the proposed device or circuit designs. Instead, I would expect quantitative comparisons that directly

demonstrate the benefits of the GMC-based in-memory computing architecture, such as improvements in energy efficiency and computational throughput, compared to a CMOS-based implementation, in order to substantiate the advantages presented in Fig. 2.

Response to 3-2:

We sincerely appreciate the reviewer's insightful comments. The issue regarding the comparison of performance between G-KAN and M-MLP in Figure 5 is crucial for enhancing the rigor and clarity of the contribution in this paper.

First, we fully understand the reviewer's concern. The performance differences between software-level algorithms (KAN and MLP) are supported by related studies. The reviewer's comments highlighted that the original phrasing might have led to a misinterpretation of Figure 5, suggesting it demonstrated "AI performance advantages brought by GMCs." We have revised the manuscript (marked in red) to clarify that the comparison aims to verify the hardware platform's ability to preserve algorithmic advantages, shifting the focus to the core contribution—performance benefits of the GMC-based in-memory computing architecture.

Next, in accordance with the reviewer's suggestion, we conducted the study on the energy efficiency of the GMC-based architecture.¹⁻⁵ Specifically, we compared the GMC-based system with two recently launched NVIDIA graphics processing units (GPUs).

The experiment was conducted on a predefined scale of the KAN [784,100,10] (grid size = 4), combining the corresponding GMC array and peripheral circuits. The

inference latency was set to 300 ns, and the average power consumption of each GMC device was measured to be 16.6345 nW. The total system energy consumption consists of the energy of the array (E_{array}) and that of the peripheral circuits (E_{perip}), with the latter primarily driven by neurons using the differential-pair mechanism, each output neuron equipped with three high-bandwidth and low-power operation amplifiers^{6,7}. The specific formulas for energy consumption calculation are as follows:

$$n_{\text{device}} = (784 \times 100 + 100 \times 10) \times 4 \times 2 = 635200 \quad (\text{R1})$$

$$n_{\text{opa}} = (100 + 10) \times 3 = 330 \quad (\text{R2})$$

$$E_{\text{array}} = 635200 \times 16.6345 \text{ nW} \times 300 \text{ ns} = 3.17 \text{ nJ} \quad (\text{R3})$$

$$E_{\text{perip}} = 330 \times 0.34 \text{ mW} \times 300 \text{ ns} = 33.66 \text{ nJ} \quad (\text{R4})$$

$$E_{\text{total}} = E_{\text{array}} + E_{\text{perip}} = 36.83 \text{ nJ} \quad (\text{R5})$$

Here, n_{device} represents the total number of devices in the GMC array, and n_{opa} denotes the total number of operational amplifiers used to implement the neurons. Moreover, according to the definition of energy efficiency^{1-3,7}, the computational workload must first be quantified. Since the GMC array performs the parameterized Gaussian transformation and accumulation ($\sum A \cdot \exp\left[-\left(\frac{x-\mu}{\sigma}\right)^2\right]$) operation in one step, such the operations need to be equated to the corresponding number of operations in CMOS architectures. In computer electronics, the operation $\sum A \cdot \exp\left[-\left(\frac{x-\mu}{\sigma}\right)^2\right]$ is executed step by step: $x - \mu \rightarrow \frac{x - \mu}{\sigma} \rightarrow \left(\frac{x - \mu}{\sigma}\right)^2 \rightarrow -\left(\frac{x - \mu}{\sigma}\right)^2 \rightarrow \exp\left[-\left(\frac{x - \mu}{\sigma}\right)^2\right] \rightarrow A \cdot \exp\left[-\left(\frac{x - \mu}{\sigma}\right)^2\right] \rightarrow \sum A \cdot \exp\left[-\left(\frac{x - \mu}{\sigma}\right)^2\right]$. Among these steps, the

transcendental function $\exp(\cdot)$ in double precision calculations involves 29 operations in floating-point architectures based on the reduction-approximation-reconstruction principle.⁸ Therefore, the $\sum A \cdot \exp\left[-\left(\frac{x-\mu}{\sigma}\right)^2\right]$ operation in computer hardware is equivalent to 35 operations in total. Based on this, the energy efficiency of the GMC-based system can be calculated as follows:

$$EE = \frac{635200 \times 35 \text{ operations}}{36.83 \text{ nJ}} = 603.64 \text{ TOPS/W} \quad (\text{R6})$$

Table R1 presents the comparison results of energy efficiency, where the energy efficiency of the GPU is calculated based on the ratio of its peak throughput to thermal design power (TDP)⁷.

Table R1 Comparison of hardware energy efficiency

Architecture	Energy Efficiency (TOPS/W)
RTX PRO 2000 Blackwell ^{*1}	7.79
RTX 2000 Ada Generation ^{*2}	2.74
This work	603.64

*1: <https://www.nvidia.com/content/dam/en-zz/Solutions/products/workstations/professional-desktop-gpus/rtx-pro-2000/workstation-datasheet-blackwell-rtx-pro-2000-nvidia-us-4016661.pdf>

*2: <https://www.nvidia.com/en-us/products/workstations/rtx-2000/>

The CIM architecture and the ability to execute one-shot transcendental functions by GMCs provide the GMC-based system with parallel computation and low power consumption, which contributes to the high energy efficiency demonstrated in Table R1. In addition, the comparison of hardware benchmark metrics has been submitted with the Supplementary Note in the revised Supplementary Information.

We would like to once again express our sincere gratitude to the reviewer for the

valuable comments and detailed guidance provided during this round of review. The reviewer's suggestions have significantly enhanced the rigor and clarity of the contribution in this paper, and the related revisions have effectively improved the final presentation of the manuscript.

3. The manuscript primarily focuses on single-device-level experiments and circuit-level design and simulations. I therefore suggest that the authors avoid using the term “implementation” in the title and clearly clarify in the main text that the proposed G-KAN has not been “implemented” at the system level to prevent overclaim of the experimental scope and ensure that the statements accurately reflect the demonstrated results.

Response to 3-3:

We sincerely thank the reviewer for this insightful suggestion. We completely agree with your concern and have revised the title accordingly. Specifically, we have replaced “Implementation” with “Architecture,” resulting in the updated title: “*Computing-in-Memory Architecture for Kolmogorov–Arnold Networks Based on Tunable Gaussian-Like Memory Cells*”.

Moreover, in both the main text and Supplementary Information, we have revised certain statements (marked in red), removing the expression that could lead to misinterpretation or over-interpretation, in order to ensure the rigor of the content and present the research more objectively. Thank you again for helping us present our work with appropriate precision.

References

- 1 Aguirre, F. *et al.* Hardware implementation of memristor-based artificial neural networks. *Nat. Commun.* **15**, 1974 (2024).
- 2 Yao, P. *et al.* Fully hardware-implemented memristor convolutional neural network. *Nature* **577**, 641-646 (2020).
- 3 Zhang, W. *et al.* Edge learning using a fully integrated neuro-inspired memristor chip. *Science* **381**, 1205-1211 (2023).
- 4 Shafiee, A. *et al.* ISAAC: A convolutional neural network accelerator with in-situ analog arithmetic in crossbars. *ACM SIGARCH Comput. Archit. News* **44**, 14-26 (2016).
- 5 Peng, X., Huang, S., Luo, Y., Sun, X. & Yu, S. DNN+ NeuroSim: An end-to-end benchmarking framework for compute-in-memory accelerators with versatile device technologies. In *2019 IEEE International Electron Devices Meeting (IEDM)*, 32.5.1-32.5.4 (IEEE, 2019).
- 6 Atef, A., Atef, M., Abbas, M. & Khaled, E. E. M. High-sensitivity regulated inverter cascode transimpedance amplifier for near infrared spectroscopy. In *2016 Fourth International Japan-Egypt Conference on Electronics, Communications and Computers (JEC-ECC)*, 99-102 (IEEE, 2016).
- 7 Lin, P. *et al.* Three-dimensional memristor circuits as complex neural networks. *Nat. Electron.* **3**, 225-232 (2020).
- 8 Harrison, J., Kubaska, T., Story, S. & Tang, P. The Computation of Transcendental Functions on the IA-64. *Intel Technology Journal Q4*, 50-56 (1999).

REVIEWER COMMENTS

Reviewer #3 (Remarks to the Author):

The authors have addressed my previous concerns. One additional comment is that the authors should clearly state that the energy-efficiency estimation added in the revised manuscript is based on idealized assumptions, as the analysis considers only the G-KAN implementation, whereas the GPUs are general-purpose computing platforms.

Response:

We are very happy to hear that you acknowledge our previous revision as effectively addressing your concerns, which also means that we have successfully refined an important detail in this study. Meanwhile, regarding your suggestion for additional clarification, we fully agree and have added the corresponding statement in the latest version of the Supplementary Note, emphasizing that the energy efficiency results are estimated based on idealized assumptions. Once again, we sincerely appreciate your valuable guidance and contribution in this work!